# The first Met Office Unified Model/JULES Regional Atmosphere and Land configuration, RAL1

Mike Bush[1], Tom Allen[1], Caroline Bain[1], Ian Boutle[1], John Edwards[1], Anke Finnenkoetter[1], Charmaine Franklin[2], Kirsty Hanley[1], Humphrey Lean[1], Adrian Lock[1], James Manners[1], Marion Mittermaier[1], Cyril Morcrette[1], Rachel North[1], Jon Petch[1], Chris Short[1], Simon Vosper[1], David Walters[1], Stuart Webster[1], Mark Weeks[1], Jonathan Wilkinson[1], Nigel Wood[1], and Mohamed Zerroukat[1]

[1]Met Office, FitzRoy Road, Exeter, EX1 3PB, UK
[2]Bureau of Meteorology (BoM), Melbourne, Victoria, Australia

**Correspondence:** Mike Bush (mike.bush@metoffice.gov.uk)

**Abstract.** In this paper we define the first "Regional Atmosphere and Land" (RAL) science configuration for kilometre scale modelling using the Unified Model (UM) as the basis for the atmosphere and the Joint UK Land Environment Simulator (JULES) for the land. "RAL1" defines the science configuration of the dynamics and physics schemes of the atmosphere and land. This configuration will provide a model baseline for any future weather or climate model developments to be described against and it is the intention that from this point forward significant changes to the system will be documented in literature. This is reproducing the process used for global configurations of the UM which was first documented as a science configuration in 2011. While it is our goal to have a single defined configuration of the model that performs effectively in all regions, this has not yet been possible. Currently we define two sub-releases, one for mid-latitudes (RAL1-M) and one for tropical regions (RAL1-T). The differences between RAL1-M and RAL1-T are documented and where appropriate, we define how the model configuration relates to the corresponding configuration of the global forecasting model.

## 1 Introduction

It is becoming standard practice for National Met Services (NMS) and those involved in the prediction of high-impact weather to use regional atmospheric and land models with grid-lengths of the order of a kilometre as their prediction systems (e.g. Baldauf et al. (2011); Brousseau et al. (2016); Bengtsson et al. (2017); Klasa et al. (2018)). While not truly resolving deep convection, kilometre scale atmospheric models are able to explicitly represent deep convective processes within the resolved dynamics. These models provide valuable information on local weather and high-impact weather that is critical to the core function of NMSs. The representation of convective systems, topographically driven weather and various mesoscale features

are generally improved with these regional modelling systems (Clark et al., 2016). In addition to weather forecasting, kilometre scale simulations are now emerging as a tool for climate projections (e.g. Kendon et al. (2017)). While there is significant computational cost to running regional models with grid-length of order kilometre scale for the many long duration runs needed for climate projections, the value of the far improved representation of weather systems, especially those related to high-impact weather, makes the computational costs worthwhile.

Over the United Kingdom, the Met Office's primary operational deterministic numerical weather prediction (NWP) weather forecast system (the UKV, Tang et al. (2013)) and ensemble prediction system (MOGREPS-UK, Hagelin et al. (2017)) are run with grid-lengths of order of a kilometre. These systems both use the Met Office Unified Model (UM, Brown et al. (2012)) as the basis for the atmosphere and the Joint UK Land Environment Simulator (JULES, (Best et al., 2011; Clark et al., 2011)) for the land. They are run in variable resolution mode, with horizontal grid-lengths in the central regions of their domains of 1.5km and 2.2km respectively. In addition, the Met Office also carry out regional kilometre scale simulations for climate projection, the latest of which have been run with horizontal grid-lengths of 1.5km over a domain covering the Southern UK (Kendon et al. (2014)), 2.2km over Europe (Berthou et al. (2018)) and 4.4km over Africa (Stratton et al. (2018)). The exact choice of grid-length and domain size is a pragmatic one where the aim is to have as good a resolution as possible while allowing the forecasts or climate projections to run in the allotted time on the computer systems available.

Regional modelling in the Met Office is not confined to the UK for weather or climate. For several international collaborations, and to meet various commitments, the Met Office also runs kilometre scale UM simulations in many other regions around the world. In addition, as part of the UM partnership, a range of institutions beyond the Met Office also run the regional model in their areas of interest. With the many regions and many users of the model it has become more important than ever to coordinate its development and have clearly defined science configurations. In this paper we define the first "Regional Atmosphere and Land" (RAL) science configuration for kilometre scale modelling using the UM and JULES. "RAL1" defines the science configuration of the dynamics and physics schemes of the atmosphere and land. This configuration will provide a model baseline for any future weather or climate model developments to be described against. It is the intention that from this point forward significant changes to the system will be documented in literature. This is reproducing the process used for global configurations of the UM which was first documented as a science configuration in 2011 (Walters et al. (2011)).

While it is our goal to have a single defined configuration of the model that performs effectively in all regions, this has not yet been possible. Currently we define two sub-releases, one for mid-latitudes (RAL1-M) and one for tropical regions (RAL1-T). The differences between RAL1-M and RAL1-T are clearly documented within this paper. Also, where appropriate, we define how the model configuration relates to the corresponding configuration of the global forecasting model defined in Walters et al. (2017).

Prior to the existence of RAL1, there was no single definition for the configuration of the regional UM model. As RAL1 is the first formally documented model configuration there is no previous baseline against which to document performance and recent developments. However, it is a goal of this paper to highlight the most recent updates and describe how these have improved performance over previous versions of the regional UM system. To do this we focus on the UK and describe the model changes against the previous operational weather prediction system. This baseline, known in the Met Office as Operational Suite 37

(OS37) was the operational system from 15th March 2016 to 08th November 2016 and will be referred to in this paper as RAL0.

In Sect. 2, we document the RAL0 configuration. In Sect. 3, we highlight the RAL1-M developments which are added to the RAL0 baseline to define RAL1-M. In Sect. 4 we document the Tropical version RAL1-T and in Sect. 5 we evaluate the performance of RAL1-M and RAL1-T configurations in five parts of the world with different meteorology. Finally, in Sect. 6 we provide some concluding remarks.

## 2 Defining Regional Atmosphere and Land - version 0 (RAL0)

### 2.1 Dynamical core: Spatial aspects

The primary atmospheric prognostics are the three-dimensional wind components, virtual dry potential temperature, Exner pressure, dry density, five moist prognostics (mixing ratios of water vapour, liquid, ice, rain and graupel) and murk aerosol (operational UK forecasts only). These prognostic fields are discretised horizontally onto a rotated longitude/latitude grid with the pole rotated so that the grid's equator runs through the centre of the model domain. Optionally, the horizontal grid may be specified as being of variable resolution, where the grid size varies smoothly from coarser resolution at the outer boundaries to a uniform fine resolution in the interior of the domain as described in Tang et al. (2013). The prognostic variables are stored using Arakawa C-grid staggering (Arakawa and Lamb, 1977) in the horizontal and Charney-Phillips staggering (Charney and Phillips, 1953) in the vertical. A terrain-following hybrid height coordinate is used that it is a mix of both pure height (i.e. flat levels) and terrain following levels (Davies et al., 2005).

In the vertical, RAL0 uses a 70 level vertical level set labelled $L70(61_t, 9_s)_{40}$, which has 61 levels below $18\,\mathrm{km}$, 9 levels above this and a fixed model lid $40\,\mathrm{km}$ above sea level. This naming convention was originally devised for global model simulations to denote the maximum number of levels that could be in the troposphere at its maximum depth of around $18\,\mathrm{km}$ ($_t$) and the number above this that would always be in the stratosphere or above ($_s$). As the mid-latitude tropopause is typically at a height of roughly 9-11 km, this level set concentrates its levels below $9\,\mathrm{km}$ with only 20 of its 70 levels above this.

### 2.2 Dynamical core: Spatio-temporal discretisation

RAL0 uses the UM's ENDGame dynamical core; a semi-implicit (SI) semi-Lagrangian (SL) formulation that solves the non-hydrostatic, fully-compressible deep-atmosphere equations of motion (Wood et al., 2014). The discrete equations are solved using a nested iterative structure for each atmospheric time step within which some terms are lagged and computed in an outer loop, while others are treated quasi-fully implicitly in an inner loop.

The SL departure point equations are solved within the outer loop using a centred average of the previous time step (time level $n$) wind and the latest estimates for the current time step (time level $(n+1)$) wind. Appropriate fields are then interpolated to the departure points, using Lagrange interpolation, with various polynomial degree options. Since pointwise Lagrange interpolation is not a conservative operation, the mass of dry air, the various water species, and any other transported tracers can drift due

to numerical errors as well as the net fluxes through the lateral boundaries. The lack of enforcement of the correct budget of such fields in RAL0 is the motivation for a change in RAL1 to use of the Zero Lateral Flux (ZLF) scheme of Zerroukat and Shipway (2017) which is outlined in section 3.1.

5 Within the inner loop, a linear Helmholtz problem is solved to obtain the pressure increment in which the Coriolis, orographic and non-linear terms are evaluated as source terms to this equation: they are averaged in an off-centred semi-implicit fashion along the semi-Lagrangian trajectory using both the known state at time level $n$ and the latest estimated (iterated) values of the fields at time level $(n+1)$. Having solved the Helmholtz problem, the other prognostic variables are obtained from the pressure increment via a back-substitution process (see Wood et al. (2014) for further details). An off-centring of 0.55 is used for all

10 variables (where a value of 0.5 represents a centred scheme and a value of 1.0 would be a fully implicit scheme).

Imposing the lateral boundary conditions (LBCs) within the solution procedure requires special treatment and details of this are given in section 8.

The physical parametrisations are split into slow processes (radiation and microphysics) and fast processes (atmospheric boundary layer turbulence, cloud and surface coupling). The slow processes are treated in parallel and computed using only

15 the previous time level $n$ model's state. They are computed once per time step before the outer loop. The source terms from the slow processes are then added explicitly to the appropriate fields before the semi-Lagrangian advection (i.e interpolation). The fast processes are treated sequentially and are computed in the outer loop using the latest estimate for the model state at the current time step, or time level $(n+1)$ (i.e., fast process are treated approximately fully-implicitly as the final state $(n+1)$ cannot be known until the end of the iteration process). A summary of the atmospheric time step is given in Algorithm 1 in

20 section 8. In practice two iterations are used for each of the outer and inner loops so that the Helmholtz problem is solved four times per time step. Finally, Table 1 contains the typical length of time step used for a range of horizontal resolutions.

There are a number of differences between the Limited Area Model (LAM) formulation of ENDGame and the global version described in Wood et al. (2014). An important one of these arises due to the iterative nature of the ENDGame algorithm and the requirement, in practice, to apply LBCs over the area covered by $\Omega_2$ and $\Omega_3$ in Figure 1. Algorithm 1 gives an outline of a typical ENDGame time step with the primary difference being the addition of the expected updating of the LAM LBCs at the end of each time step but also the addition of an *update dynamics only* LBCs step during the main iteration. The main purpose of this step is to reset the new time level's velocities to be compatible with the LBCs since these will have been altered in the Helmholtz/inner loop section.

**Table 1.** Typical time step for a range of horizontal resolutions.

| Radial resolution | Nominal physical resolution | Typical time step |
| --- | --- | --- |
| 0.0135° | 1.5 km | 60 s |
| 0.02° | 2.2 km | 100 s |
| 0.04° | 4.4 km | 100 s |

## 2.3 Lateral Boundary Conditions (LBCs)

LAMs solve the atmospheric equations on a physical domain $\Omega_1$ subject to LBCs provided by a driving (generally a global) model, imposed on the periphery of $\Omega_1$ (see Figure 1). The UM's treatment of LBCs uses the method of relaxation/blending (Davies, 1976; Perkey and Kreitzberg, 1976). The relaxation method requires the LBCs to be a data region (shown in Figure 1 by the RIM region $\Omega_2 + \Omega_3$) with several grid points so that the driving model (or LBCs) and the LAM solutions are gradually blended to reduce wave reflections from the boundaries (Marbaix et al., 2003). Additionally, for SL models the LBCs are further extended, as a fluid parcel ending up inside the domain $\Omega_1$ may have come from a region outside $\Omega_1$ and far away from its boundary $\Gamma_1$ depending on the scale of the horizontal wind and the size of the time step used. The number of points defines the size of the LBCs and depends on the order of interpolation used for SL advection, the size of the blending zone and the maximum (expected) Courant number allowed (Aranami et al., 2014). The UKV model uses $\Omega_2 = 3$, $\Omega_3 = 5$ and $\Omega_4 = 7$.

The solver is identical in structure between LAM and global with the application of the boundary conditions on the Helmholtz equation being the main difference. The pressure boundary condition is of Dirichlet type with the (hydrostatically balanced) LBC held fixed on the outer most part of $\Omega_3$. LBC vertical velocity is assumed to be zero while that obtained from the inner loop will be non-zero. An implicit vertical damping profile is employed whose damping rate is proportional to the blending weights used in regions $\Omega_2$ and $\Omega_3$. Not only does this help with the model imbalance it also reduces the iteration count of the linear solver while also improving model stability.

Another difference between the LAM models and global is the calculation of trajectories (departure points) for the SL transport. The absence of the polar singularity allows for a much simpler (less computationally expensive) departure point algorithm, compared to Thuburn and White (2013), and is essentially described in Allen and Zerroukat (2016) but with the additional constraint of the departure points being clipped to $\Omega_3$ in Figure 1. At excessively large Courant numbers, which can occur sporadically when the jet stream intersects the lid of the model, there is the potential for the data required to interpolate the fields to be off-processor. The solution is derived from observing that for a halo width $H$ and for cubic Lagrange interpolation, the largest westward Courant number allowable is $H - 1$ while the largest eastward Courant number is $H - 2$ and similarly for North and South. This observation allows for the introduction of a *trajectory* clipping algorithm which looks at the distance of the departure point (in grid point space) from the arrival point and moves it, depending on the direction of the flow, if the distance is greater than the maximum allowable to the furthest grid point at which there would be no issues. At points that have been moved the interpolation weights are reset to $0.5$ to remove any potential biases. Note that, because this calculation is performed in grid point space, the variation of the Courant number with the variable grid resolution is automatically accounted for.

## 2.4 Solar and terrestrial radiation

Shortwave (SW) radiation from the Sun is absorbed and reflected in the atmosphere and at the Earth's surface and provides energy to drive the atmospheric circulation. Longwave (LW) radiation is emitted from the planet and interacts with the atmosphere, redistributing heat, before being emitted into space. These processes are parametrised via the radiation scheme,

which provides prognostic atmospheric temperature increments, prognostic surface fluxes and additional diagnostic fluxes. The SOCRATES [1] radiative transfer scheme (Edwards and Slingo, 1996; Manners et al., 2018) is used with a configuration based on GA3.1 (Walters et al., 2011). Solar radiation is treated in 6 SW bands and thermal radiation in 9 LW bands. In the LW an approximate treatment of scattering is used (Manners et al., 2018) to reduce execution time.

Gaseous absorption uses the correlated-$k$ method with coefficients identical to the GA3.1 configuration. Twenty-one (21) $k$ terms are used for the major gases in the SW bands, with absorption by water vapour ($H_2O$), carbon dioxide ($CO_2$), ozone ($O_3$), and oxygen ($O_2$). Thirty-three (33) $k$ terms are used for the major gases in the LW bands, with absorption by $H_2O$, $O_3$, $CO_2$, $CH_4$, $N_2O$, CFC-11 ($CCl_3F$) and CFC-12 ($CCl_2F_2$). Of the major gases considered, only $H_2O$ is prognostic; $O_3$ uses a climatology, whilst other gases are prescribed using a fixed mass mixing ratios and assumed to be well mixed.

Absorption and scattering by aerosols is included based on a simple climatology of five species: water soluble, dust, oceanic, soot and stratospheric aerosols. The component in the planetary boundary layer is distributed over approximately 3.2km of the atmosphere (lowest 30 model levels) and the contribution from dust has been scaled by 0.3333 compared to the original climatology of Cusack et al. (1998) as the dust loading of the basic climatology over land (which includes arid areas) is too high for the UK.

The parametrisation of cloud droplets is described in Edwards and Slingo (1996) using the method of "thick averaging". Padé fits are used for the variation with effective radius, which is computed from the number of cloud droplets calculated in the microphysics scheme (see section 2.5). The parametrisation of ice crystals is described in Baran et al. (2016).

The sub-grid cloud structure is represented using separate cloud fractions for the liquid and ice components with the liquid water mass mixing ratio scaled by a factor of 0.7 to represent the effect of cloud inhomogeneity as described in Cahalan et al. (1994). Cloud fractions in adjacent layers in the vertical are maximally overlapped, while clouds separated by clear-sky are randomly overlapped. Full radiation calculations are made every $15\,\mathrm{min}$ using the instantaneous cloud fields and a mean solar zenith angle for the following $15\,\mathrm{min}$ period. Corrections for the change in solar zenith angle on every model time step and the change in cloud fields every $5\,\mathrm{min}$ are made as described in Manners et al. (2009).

The emissivity and the albedo of the surface are set by the JULES land surface model (see section 2.8). A single frequency-averaged emissivity is specified for each surface type (see Walters et al. (2014) for the numerical values). For the surface albedo, the radiative transfer in plant canopies uses the two-stream radiation scheme and spectral parameters of Sellers (1985).

The direct SW flux at the surface is corrected for the angle and aspect of the topographic slope and for shading by surrounding terrain. The net LW flux at the surface is corrected for the resolved sky-view factor due to the surrounding terrain (Manners et al., 2012).

## 2.5 Microphysics

The formation and evolution of precipitation due to grid scale processes is the responsibility of the microphysics scheme. The microphysics scheme has prognostic input fields of temperature, moisture, cloud and precipitation from the end of the previous time step, which it modifies in turn. The microphysics used is a single moment scheme based on Wilson and Ballard (1999),

---

[1] https://code.metoffice.gov.uk/trac/socrates

with extensive modifications. We make use of prognostic rain, which allows three-dimensional advection of the rain mass mixing ratio. This has been shown to improve precipitation distributions over and around mountainous regions, especially with the smaller grid spacings used in the RAL configurations (Lean et al., 2008; Lean and Browning, 2013). Prognostic graupel has also been included, this allows for the explicit representation of a second, more dense ice category which is useful for hail forecasting at kilometre scale resolutions as well as being a prerequisite for lightning forecasting (Wilkinson and Bornemann, 2014).

The warm-rain scheme is based on Boutle et al. (2014b), and includes an explicit representation of the affect of sub-grid variability on autoconversion and accretion rates (Boutle et al., 2014a). We use the rain-rate dependent particle size distribution of Abel and Boutle (2012) and fall velocities of Abel and Shipway (2007), which combine to allow a better representation of the sedimentation and evaporation of small droplets. The cloud droplet number concentration can be determined from assuming either a) a fixed climatological aerosol, or b) using a single-species prognostic aerosol which has been developed for forecasts of visibility (Clark et al., 2008). For the cases where single-species prognostic aerosol is used, the aerosol concentrations are coupled to the cloud drop number using the methodology described in Wilkinson et al. (2013) and modified following Osborne et al. (2014). In the case of the fixed climatological aerosol, the parametrisation of Jones et al. (1994) is used. In both cases, droplet numbers are reduced near the surface for effective fog simulation and changes included in RAL1 are described in section 3.3).

Ice cloud parametrisations use the generic size distribution of Field et al. (2007) and mass-diameter relations of Cotton et al. (2013). The fall speed of ice used is the dual fall-speed as described in Furtado et al. (2015), where the lowest value of two computed fall speed relations is used. This represents the fact that the Field et al. (2007) parametrisation includes contributions from both smaller ice crystals and larger ice aggregates.

Unlike the GA configurations, there is no requirement for multiple sub-time stepping of the microphysics scheme as the model time step in the RAL configurations is shorter than the 2-minute period used as a sub-time step in the GA configurations.

As in Stratton et al. (2018), the output taken immediately after the microphysics scheme drives a lightning parametrisation, based on McCaul et al. (2009); with the discharge of lightning flashes in the column being determined as described in Appendix A of Wilkinson (2017). This has been shown to be of benefit for a high-profile event (Wilkinson and Bornemann, 2014) and to perform well during the summer months (Wilkinson, 2017).

## 2.6 Large-scale cloud

Due to sub-grid inhomogeneity, clouds will form well before the humidity averaged over the size of a grid-box reaches saturation and this is still true when grid-box size is at the kilometer scale (Boutle et al., 2016). A cloud parametrisation scheme is therefore required to determine the fraction of the grid-box which is covered by cloud and the amount and phase of condensed water contained in those clouds. The formation of clouds will convert water vapour to liquid or ice and release latent heat. The cloud cover and liquid and ice water contents are then used by the radiation scheme to calculate the radiative impact of the clouds and by the microphysics scheme to calculate whether any precipitation has formed.

RAL0 uses the Smith (1990) cloud scheme. This is a diagnostic scheme, in which the cloud cover is calculated only from information available at that moment in time. The scheme relies on a definition of critical relative humidity, RHcrit, which is the grid-box mean relative humidity at which clouds start to appear. The value of RHcrit is set to 0.96 at the surface and decreases monotonically to 0.80 at 850m (model level 15). It is then held fixed above that.

For liquid cloud, the Smith cloud scheme is built around an assumption that sub-grid temperature and humidity fluctuations can be described by a symmetric triangular probability distribution function (PDF). One consequence of this PDF assumption is that the grid-box has 50% cloud cover when the total relative humidity, RHt=(qv+qcl)/qsat (where qv is the vapour, qcl is the liquid content and qsat is the saturation specific humidities), reaches 100% and that the grid-box only becomes overcast when RHt>=2-RHcrit. However, observations such as Wood and Field (2000) suggest that the cloud fraction should be larger than 0.5 when RHt=100%. As a result, an empirically-adjusted cloud fraction (EACF) is used with the Smith scheme in kilometre-scale models. The relative humidity at which cloud first appears is unchanged, but the smooth function linking cloud fraction to relative humidity increases more rapidly so that cloud fraction is 0.70 when RHt=100%.

Forecasts using the EACF still under-estimate cloudiness however, especially the thin clouds forming below a temperature inversion that do not fill the entire depth of a model layer. So an area cloud fraction scheme is also used, which follows a similar approach to that described by Boutle and Morcrette (2010). Each model level is split into three and vertical interpolation is used to find the thermodynamic values in the sub-layers. However, if there is a strong gradient in RH, due to the presence of a capping inversion, the thermodynamic properties of the sub-layer are found by extrapolation from above and below instead. This sharpens the inversion and can increase the RH in the sub-layers below it. The Smith cloud scheme, itself modified to use the EACF, is then called on each of the 3 sub-layers. The cloud fraction for use by the microphysics is set to the mean of the cloud fractions over the 3 sub-layers, while the cloud fraction seen by radiation is set the maximum of the values from the 3 sublayers.

The ice cloud fraction is parametrised as described by Abel et al. (2017) where it is diagnosed from the ice water content.

## 2.7 Atmospheric boundary layer

The parametrisation of turbulent motions in kilometre scale models requires special treatment because, although most turbulent motions are still unresolved, the largest scales can be of a similar size to the grid-length. The model must therefore be able to parametrize the smaller scales, resolve the largest ones if possible, and not alias turbulent motions smaller than the grid-scale onto the grid-scale. The "blended" boundary-layer parametrisation described by Boutle et al. (2014b) is used to achieve this. This scheme transitions from the 1D vertical turbulent mixing scheme of Lock et al. (2000), suitable for low-resolution simulations such as GA configurations, to a 3D turbulent mixing scheme based on Smagorinsky (1963) and suitable for high-resolution simulations, based on the ratio of the grid-length to a turbulent length scale. The blended eddy diffusivity, including any non-local contribution from the Lock et al. (2000) scheme, is applied to down-gradient mixing in all three dimensions, whilst appropriately weighted non-local fluxes of heat and momentum are retained in the vertical for unstable boundary-layers. The configuration of the Lock et al. (2000) scheme is similar to that of GA7 (Walters et al., 2017), with differences as follows: (i) for stable boundary layers, the "sharp" function is used everywhere, but with a parametrisation of sub-grid drainage

5   flows dependent on the sub-grid orography (Lock, 2012), (ii) heating generated by frictional dissipation of turbulence is not represented, and (iii) the parametrisation of shear generated turbulence extending into cumulus layers (Bodas-Salcedo et al., 2012) is not used.

The functions that are used to include the effects of stability on turbulence, via the Richardson number ($Ri$), follow Brown (1999):

$$f_m = (1 - c_{LEM} Ri)^{1/2} \qquad\qquad f_h = \frac{1}{Pr_N} (1 - b_{LEM} Ri)^{1/2} \qquad\qquad (1)$$

where $Pr_N$ is the neutral Prandtl number ($= 0.7$). In RAL0, the constants $b_{LEM}$ and $c_{LEM}$ are both equal to 1.43 (which gives Brown's "conventional" model). RAL0 uses a mixing length that is a fraction (0.15) of the depth of any layer where $Ri$ is less than a critical value ($Ri_{crit} = 0.25$), within that layer, or 40m if larger.

In an effort to improve the triggering of explicit convection, stochastic perturbations to temperature are applied. Designed
to represent realistic variability that might be seen due to large boundary layer eddies, the perturbation scale for potential temperature, $\theta$, is taken as $\theta_* = \overline{w'\theta'}|_s / w_m$, where $\overline{w'\theta'}|_s$ is the surface turbulent flux of $\theta$ and the turbulence velocity scale $w_m$ is given by $w_m^3 = u_*^3 + c_{ws} w_*^3$. Here $u_*$ is the friction velocity and $w_*$ the convective velocity scale, with $c_{ws} = 0.25$. Finally $\theta_*$ is constrained to be positive and less than 1 K. Loosely based on Munoz-Esparza et al. (2014), the random number field that multiplies the perturbation scale is held constant over 8 grid-length squares in the horizontal and the perturbations are
applied uniformly in the vertical up to the lower of two-thirds of the boundary layer depth and 400m.

## 2.8   Land surface and hydrology

Exchanges of mass, momentum and energy between the atmosphere and the underlying land and sea surfaces are represented using the community land surface model JULES (Best et al., 2011; Clark et al., 2011). The configuration adopted in RAL0 largely follows that of GL7.0, as described by (Walters et al., 2017). In keeping with the seamless approach to model devel-
opment, the aim is to minimize the differences between configurations, but different developmental priorities for regional and global modelling can result in differences between the configurations, even if there is no compelling scientific motivation to maintain them. We now list and explain the non-trivial differences.

Because the UKV was developed for short-range forecasting over the UK, the treatment of surface exchange over sea and sea ice has been less of a priority than in the global model, so that RAL0 is less advanced in its treatment. A fixed value of
Charnock's coefficient (0.011) is used to determine the surface roughness over open sea, as opposed to the COARE algorithm in GL7.0. GL7.0 also includes a more advanced parametrisation of the sea surface albedo (Jin et al., 2011) that incorporates a dependence on the wind speed and chlorophyll concentration. This has not yet been introduced into RAL0, which still uses an earlier scheme based on Barker and Li (1995). Similarly, because the regional model has not yet been used operationally over sea ice, several recent modifications to sea-ice parameters have not yet been introduced into the regional configuration. Sea-ice is not present in the simulations shown below, so these settings are not relevant to any results presented here.

Although both GL7.0 and RAL0 include the multilayer snow scheme, different densities of fresh snow are specified: in GL7.0 the value is 109 $kgm^{-3}$, while in RAL0 a value of 170 $kgm^{-3}$ is used as more representative of the conditions in the UK. In the future, it is hoped that it will be possible to relate the density to local meteorological conditions.

Both GL7.0 and RAL0 represent the radiative transfer in plant canopies using the two-stream radiation scheme of Sellers (1985), with the leaf-level reflection and transmission coefficients presented in that paper. However, in GL7.0, an adjustment to these parameters is made as the model runs to make the grid-box mean albedo agree more closely with a climatology derived from GlobAlbedo. While developing this adjustment for GL7.0, the simulated direct albedos were found to be unrealistic and the diffuse albedos were used for both the direct and diffuse beams. As implemented in RAL0, there is no adjustment to a climatology and both the direct and diffuse albedos are used. Further discussion of these issues may be found in section 3.5.

Two differences in soil hydrology should be noted. Whereas the more elaborate TOPMODEL scheme is used to represent soil moisture heterogeneity in GL7.0, the simpler PDM scheme is used in RAL0 (consult Best et al. (2011) for details of these schemes). Also, in RAL0, if the simulated soil moisture rises above the saturated water content, the excess is assumed to move upwards and to contribute to surface runoff. This is considered more realistic than the alternative of routing the excess moisture downwards, except in regions of partially frozen soils (Best et al., 2011). In GL7.0 the excess moisture is routed downwards.

In GL7.0 urban surfaces are represented by a single urban tile, but in RAL0 two separate tiles for street canyons and roofs are used (Porson et al., 2010). Currently the two tile scheme is limited to domains over the UK due to the availability of morphology data.

## 2.9 Lower boundary condition (ancillary files) and forcing data

In the UM, the characteristics of the lower boundary, the values of climatological fields and the distribution of natural and anthropogenic emissions are specified using ancillary files. Use of correct ancillary file inputs can potentially play as important a role in the performance of a system as the correct choice of many options in the parametrisations described above. In the future we may consider the source data and processing required to create ancillaries as part of the definition of the RAL configurations as is the case in global configurations. However we currently leave ancillaries outside the formal definition of RAL as there has been no systematic evaluation of the impact on performance of different ancillary file inputs and the existence of many country specific datasets (that are of better quality or higher resolution) mean that different applications (especially operational ones such as UKV/MOGREPS-UK) use different source datasets, sometimes even combining different datasets within the model domain. An example of this is described in section 3.5.

Table 4 in the appendix contains the main ancillaries used in RAL applications as well as references to the source data from which they are created.

## 2.10 Other differences from GA7 due to horizontal resolution

The high horizontal resolutions used for RAL simulations mean that RAL0 runs with the convection parametrisation switched off, relying on the model dynamics to explictly represent convective clouds. Although it is acknowledged that not all types of convection are represented with such grid-spacing, this choice was made in the current absence of a scale-aware convection

scheme which correctly parametrizes sub-grid convective motion and hands over to the model dynamics for clouds larger than the model filter scale. Projects are underway to develop convection schemes for use in atmospheric models at all resolutions with grid spacings $O(1\text{--}100,\text{km})$, which could be incorporated into a future RAL release.

Also, RAL0 does not include a sub-grid parametrisation scheme for either orographic or non-orographically forced gravity waves. However for those non-UK-area models that run with a grid-length of $0.04\,^{\circ}$ (4.4 km), the inclusion of the effective

roughness and gravity wave drag schemes (both as used in GA7) was found to be beneficial to near-surface verification scores.

## 3    Developments included in RAL1

This section describes the RAL1-M developments which when added to the RAL0 base define RAL1-M. The Regional Model Evaluation and Development (RMED) processes at the Met Office makes use of an online 'ticket' tracking system which allows scientists to document changes to the model. A ticket number is assigned to each model development and thus it is clear to

all developers and external collaborators which tickets are included in any one configuration. In this section we discuss the major developments to RAL1 and reference them by ticket number to inform both the development community and for future cross-reference. For ease of reference, a complete list of all the RMED tickets included in RAL1 can be found in Table 6.

### 3.1    Dynamical formulation and discretisation

**Conservative advection for moist prognostics (RMED ticket #2)**

The mass conservation for mixing-ratios is achieved with the ZLF (Zero-Lateral-Flux) scheme (Zerroukat and Shipway, 2017). This scheme is computationally efficient and exploits the relatively large width of LBCs used for semi-Lagrangian based LAMs (i.e., the size of the extra extended computational zone $\Omega_E = \Omega_2 + \Omega_3 + \Omega_4$ shown in Figure 1). Assuming that the size of $\Omega_E$ is sufficiently large (> 2 points) it can be divided into two regions as shown in Figure 1 with a dotted line/boundary $\Gamma_2$, which will be referred as the ZLF boundary. It is also very common that the wind and the time step used are such that the horizontal

Courant number in the RIM zone is smaller than half of the RIM size. Under these conditions, the SL advection solution for all the points inside the region $\{\Omega_1 + \Omega_2\}$ (which includes the forecasting zone) will be unaffected by the field beyond the ZLF boundary $\Gamma_2$. Therefore, for convenience, the advection solves a modified problem, whereby inside $\Gamma_2$ the advected quantity is the original field, whereas the field beyond $\Gamma_2$ is zeroed. This modification does not affect the solution inside the domain $\{\Omega_1 + \Omega_2\}$ and hence it is equivalent to the original problem. However, this modification allows us to impose a simple mass

conservation constraint over the whole extended computational domain $\{\Omega_1 + \Omega_2 + \Omega_3\}$ where there is no need to compute lateral fluxes because they are zero by construction (see details in Zerroukat and Shipway (2017)). This is quite an important simplification from the case where one would like to impose a mass conservation budget for the forecast/physical domain $\Omega_1$, which requires knowledge of mass fluxes through its lateral boundary $\Gamma_1$, which are complicated and computationally expensive to compute (Aranami et al., 2014). The ZLF scheme has two components: the first part (just explained above) which allows us to avoid computing expensive lateral fluxes, while the second part is the redistribution of the mass conservation error

using the optimized conservation scheme (Zerroukat and Allen, 2015). Note that the zeroing is just an intermediate temporary step used during the advection, because the zeroed-region gets overwritten by the appropriate LBC data at the end of the time step.

### 3.2  Solar and terrestrial radiation

**Improved treatment of gaseous absorption (RMED ticket #9)**

The treatment of gaseous absorption has been significantly updated to the configuration used with GA7  (Walters et al., 2017).

Forty-one (41) $k$ terms are used for the major gases in the SW bands with an improved representation of $H_2O$, $CO_2$, $O_3$, and $O_2$ absorption and the addition of absorption from nitrous oxide ($N_2O$) and methane ($CH_4$). These changes result in increased atmospheric absorption and reduced surface (clear-sky) fluxes.

Eighty-one (81) $k$ terms are used for the major gases in the LW bands with an improved representation of all gases. This

results in reduced clear-sky outgoing LW radiation, and increased downwards surface fluxes.

The method of "hybrid" scattering is used in the LW which runs full scattering calculations for 27 of the major gas $k$-terms (where their nominal optical depth is less than 10 in a mid-latitude summer atmosphere). For the remaining 54 $k$-terms (optical depth $> 10$) much cheaper non-scattering calculations are run.

In both spectral regions the band-by-band breakdown of absorption is improved which should improve interaction with

band-by-band aerosol and cloud forcing.

### 3.3  Microphysics

**Improved droplet number profile in the lower boundary layer (RMED ticket #1)**

Previous work by Wilkinson et al. (2013) had discussed a pragmatic method of reducing the cloud droplet number near the surface, often referred to as a "droplet taper". This reduction accounts for the fact that aerosol activation and cloud droplet

numbers measured in fog are often much lower than those found in more elevated clouds, despite the fact that the underlying aerosol concentrations are generally higher. Recent work by Boutle et al. (2018) utilising new observations (Price et al., 2018) has enhanced our understanding of this process, demonstrating that weak updraughts and low supersaturations in fog are the reason for the limited aerosol activation. Boutle et al. (2018) showed that even the droplet number profile of Wilkinson et al. (2013) gave values too high too close to the surface. This triggered a feedback process whereby fog became too deep and well-developed too quickly, resulting in significant errors to fog forecasts. Boutle et al. (2018) proposed a modified parametrisation for the near-surface droplet number, which was shown in forecast trials to be of significant benefit. Therefore RAL1 has adopted the droplet number parametrisation proposed by Boutle et al. (2018), i.e. droplet numbers are held at 50 cm$^{-3}$ throughout the lowest 50 m of the atmosphere, before transitioning to the cloudy values as described in Wilkinson et al. (2013). We note that

5  this is still a pragmatic choice based on model performance, and further work is required to develop an activation scheme which correctly accounts for aerosol effects and is valid in the foggy regime.

### 3.4 Atmospheric Boundary Layer

**Updates to stochastic boundary layer perturbations (RMED ticket #25)**

Several updates were made to the stochastic perturbations in the boundary layer (described in section 2.7) for RAL1 in order to further enhance the triggering of convective activity. The first was also to apply the perturbations to specific humidity, using the same formulation for the perturbation scale (based on the surface humidity flux), constraining the moisture scale to be less than 10% of the specific humidity itself. Secondly the random number field was changed from being randomly different every time step to being updated in time following McCabe et al. (2016) using a first-order auto-regression model with the auto-correlation coefficient set to give a decorrelation timescale of 600 s, an approximate eddy-turnover timescale. This temporal coherence of the perturbations results in a greater resolved scale dynamical response. Finally, in the vertical the perturbations are now scaled by a piece-wise linear "shape" function equal to unity in the middle of the boundary layer and zero at the surface and top of the sub-cloud layer and are only applied where a cumulus regime is diagnosed (see Lock et al. (2000)). These were pragmatic changes to avoid the perturbations strongly influencing the screen-level temperature diagnostic, which had been found to lead to degradation of deterministic measures of skill (such a root-mean-square error).

**Revision of free-atmospheric mixing length (RMED ticket #12)**

In RAL1-M, the free-tropospheric (i.e., above the boundary layer) mixing length is reduced everywhere to its minimum value of 40m, which was found to give better, more rapid initiation of showers in UKV than the interactive mixing length used in RAL0 (and also kept for RAL1-T, see section 4).

**Improved representation of mixing across the boundary layer top (RMED ticket #5)**

This ticket allows the boundary layer scheme's explicit entrainment parametrisation to be distributed over a vertically-resolved inversion layer, instead of always assuming the inversion to be subgrid. As a result it allows a smoother transition in the vertical between the boundary layer and free troposphere. More details are given in Walters et al. (2017) (under GA ticket #83), noting that the additional representation of "forced cumulus clouds" within a resolved inversion is only included in RAL1-T (see section 4) as that requires the PC2 cloud scheme to be used.

**Reductions in sensitivity to vertical resolution (RMED ticket #10)**

The turbulent mixing and entrainment in cloud-capped boundary layers in the Lock et al. (2000) scheme is parametrized in terms of (among other things) the strength of cloud-top radiative cooling. This is calculated by differencing the radiative flux across the top grid-levels of the cloud layer. The complexity of the calculation is increased by making allowance for changes in the height of cloud between radiation calculations (which are not performed on every model time step for reasons of computational efficiency). A new methodology is introduced that identifies where the LW radiative cooling profile transitions from free-tropospheric rates above the cloud to stronger rates within it. It has very little impact at current vertical resolutions

(typically greater than 100m) but has been demonstrated in the single column version of the model to be robustly resolution independent down to grids of only a few metres.

## 3.5 Land surface and hydrology

**Improvements to land usage and vegetation properties (RMED ticket #3)**

There are four changes to the representation of the land surface in RAL1:

1. updated land use mappings, mainly removing small (<0.2) bare soil tile fractions from land use categories such as grassland. For UK areas the non-UK source data is changed from IGBP to CCI (for more details, see table 4). For operational UKV purposes, though, the IGBP land mask is retained to reduce downstream impacts.

2. reduction in the bare soil fraction of short vegetation tiles (given by $F = e^{-kext*LAI}$, where $LAI$ is the leaf area index) by increasing $k_{ext}$ from 0.5 to 1.

3. reduction of the scalar roughness lengths for the grass tiles, by reducing its ratio to the momentum roughness from 0.1 to 0.01. This enhances the difference between skin and near-surface air temperatures.

4. modifications to the canopy radiation model. Two modifications were made in the canopy radiation model. The treatment of direct solar radiation described in Sellers (1985) and originally implemented in JULES applies only in the case of isotropic scattering. It was therefore extended in RAL1 to account for non-isotropic scattering of direct radiation. Following the assessment of Lawrence et al. (2011), using the CLM4 model, that the leaf-level near-infrared reflection coefficients given by Sellers (1985) for grass are too high, the leaf-level transmission and reflection coefficients for all plant canopies were reviewed and modified. The main effect of these changes was to reduce the near-infrared albedo of short vegetation, thus increasing daytime temperatures.

The most significant of all these changes is the increase in vegetation cover at the expense of bare soil, which is a combination of (1) and (2). This provides more insulation between the the atmosphere and the underlying soil which results in more rapid evolution of surface and near-surface air temperatures, especially across the diunral cycle, and a reduction in the diurnal temperature range of the upper soil levels. Both are found to give improved agreement with in-situ observations.

## 4 The tropical configuration RAL1-T

In sections 2 and 3 we have described the mid-latitude sub-version of RAL1. In this section we describe the tropical sub-version of RAL1 known as RAL1-T. Ideally we would prefer to have one configuration for use anywhere in the world and this is an aspiration for the future. With current parameterisations, however, we find we need two configurations to get good performance in the two different areas.

One of the major reasons why we need two configurations is that convection is sometimes very under-resolved in the UK in km scale models, particularly in cases of small, shallow showers. This can mainifest itself as small showers initiating too late or

not at all. In order to cope with this, RAL1-M, has relatively weak turbulent mixing and stochastic perturbations to encourage the model fields to be less uniform and help convection initiate. If the model is run with these in the tropics the model initiates too early and convective cells tend to be too small.

## 10 Representation of turbulence (RMED tickets #12 and #26) and BL stochastic perturbations (RMED ticket #25)

There are two differences in the representation of turbulence between RAL1-M and RAL1-T, namely in the form of the stability functions and in the free-atmospheric mixing length. Both give enhanced turbulent mixing in RAL1-T compared to RAL1-M. RAL1-T uses the Brown (1999) "standard" model whilst RAL1-M uses the Brown (1999) "conventional" model. RAL1-T retains RAL0's interactive free-atmospheric mixing length, whilst RAL1-M uses a value of 40m. The other related change is that RAL1-T does not use the stochastic boundary layer perturbations. For more details and a summary of differences between RAL1-T and RAL1-M, see Table 2.

### Improvements to cloud scheme (RMED ticket #16)

RAL1-T has three extra prognostic fields (liquid fraction, ice fraction and mixed-phase fraction) as it uses the prognostic cloud prognostic condensate (PC2) cloud scheme (Wilson et al., 2008a). PC2 calculates sources and sinks of cloud cover and condensate and advects the updated cloud fields, hence adding some memory into the system. One advantage of PC2 over the Smith schemes is the looser coupling between variables, hence allowing a cloud to deplete its liquid water content while maintaining high cloud cover. The PC2 scheme performs better than the Smith scheme in climate simulations (Wilson et al., 2008b) and for global numerical weather prediction (Morcrette et al., 2012). It is worth noting that when run in a model using a convection scheme, the detrainment of cloud from convection is a key source of cloudiness (Morcrette and Petch, 2010; Morcrette, 2012b). When run in a model without a convection scheme (such as the RAL configuration), cloud formation from convective motions will be represented by a combination of PC2 initialization (near convective cloud base), followed by PC2 pressure forcing through the rest of the updraught. In the PC2 scheme, cloud erosion is a process that accounts for evaporation and reduction of cloud cover due to unresolved mixing near cloud edges. In the original implementation of PC2 (Wilson et al., 2008a) erosion was carried out as part of the call to the convection scheme, but in RAL1, which has no call to the convection scheme, the erosion process has been moved to occur within the microphyics scheme. In RAL1-T, the PC2 scheme is implemented as in the GA7 global model configuration (Walters et al., 2017). That is, the formulation of cloud erosion accounts for the apparent randomness of cloud fields, as described in Morcrette (2012a) and the RHcrit is calculated from the turbulent kinetic energy (Van Weverberg et al., 2016).

Another difference, particularly affecting convection in the tropics, is that the tropopause is deeper than in mid-latitudes. In order to take account of this RAL1-T uses a vertical level set labelled L80(59t;21s)38:5, which adds some additional vertical resolution in the tropical upper-troposphere at the expense of resolution in the lower boundary layer.

15 Figure 2 illustrates the above discussion by showing the effect of running RAL1-M and T for a case of small showers in the U.K. Unlike RAL1-M, when compared to the radar RAL1-T initiates too late and produces too large and too few showers.

**Table 2.** RAL1-M and RAL1-T differences. %

| RMED Ticket | Science difference | RAL1-M | RAL1-T |
|---|---|---|---|
| 12 | BL Free Atmospheric mixing length | 40m | interactive mixing length |
| 26 | BL Stability functions | $b_{LEM}$ =1.43, $c_{LEM}$ =1.43 | $b_{LEM}$ =40, $c_{LEM}$ =16 |
| 25 | BL stochastic perturbations to temperature and moisture | on (improved triggering) | off |
| 16 | Cloud Scheme | Smith (diagnostic) | PC2 (prognostic) |

## 5 Model evaluation

In this section we apply a range of evaluation methods to demonstrate the performance of RAL1. The regional model evaluation process is rapidly evolving and has already benefitted from the multi-institutional UM partnership. The regional model is run by UM partners in a variety of domains worldwide and RAL1 marks a baseline to which all centres can now focus future evaluation effort.

In this first documentation of the regional model we have focused on performance of RAL1 over the UK, Singapore, Australia, the Western North Pacific (Philippine Area of Responsibility for Tropical Cyclone forecasting) and the USA. This allows inspection of model behaviour in a variety of climatic zones and for different weather phenomena.

A range of evaluation methods are required to assess the performance of models. Verification skill scores, anomaly plots and case studies all provide useful information which builds a picture of model characteristics and skill. Kilometre scale models behave and look differently to models where the convection is parameterised. Convection in these models is more likely to look realistic than in a global (parameterised) model and may mimic many of the characteristics seen in satellite images and animations. However, although the detail looks realistic, it may not always be skilful. It is a challenge to create metrics which can truthfully represent the benefit of kilometre scale models as well as clarify their limitations.

Mittermaier (2014) proposed a new spatial and inherently probabilistic framework for evaluating kilometre scale models and Mittermaier and Csima (2017) provide a historical overview of performance of the 1.5 km model using this new "High Resolution Assessment" (HiRA) framework. The framework uses synoptic observations, but instead of using the single nearest model grid point, it uses a neighbourhood of model grid points centered on the observing location to acknowledge the fact that added detail may not be in the right place at the right time. These points can be treated as a pseudo ensemble, and we can compute ensemble metrics as it can be assumed that all the forecast values in the neighbourhood are equally likely outcomes at the observing location. One caveat to ensure this assumption holds is that the neighbourhood must not be too large. The framework can be applied to deterministic and ensemble forecasts, including the control member of the ensemble. Whilst it may be less than intuitive to think that a forecast neighbourhood is required for temperature, it was shown in Mittermaier and Csima (2017) that all variables benefited from the use of at least a 3 x 3 neighbourhood, but that too large neighbourhoods may be detrimental for some variables, including temperature. The HiRA Ranked Probability Score (RPS) is used for non-normally distributed or spatially discrete variables whilst the Continuous Ranked Probability Score (CRPS) is used for temperature.

The Fractions Skill Score (FSS, Roberts and Lean, 2008) requires spatial observation-based analysis. Over the UK this is a radar-based analysis, though more recently a GPM based product (Skofronick-Jackson et al., 2017) has also been used for evaluating kilometre grid scale configurations in the tropics. Analyses based on remote-sensed data and may not be accurate in an absolute sense (no observations are perfect and error-free). The FSS is sensitive to the bias (Mittermaier and Roberts, 2010), and for this reason the FSS is generally used in conjunction with percentile thresholds, where all the values in the forecast and analysis domains are ranked separately, and the physical value associated with a specific centile is extracted. This quantile transformation removes the bias so that the FSS based on percentile thresholds offers a measure of field texture, pattern and areal extent, and *not* intensity.

## 5.1 Introducing the RMED "toolbox"

To assist the RMED processes, an evaluation toolbox has been created to support model development. The main purpose is to ensure a uniformity of verification and diagnostic output across multiple users and institutions. Version 1 of the toolbox was released in time for the RAL1 assessment. It contains a selection of verification techniques and diagnostic tools, intent on enabling the comparison with point observations as well as gridded truth sources. One of the outputs of the toolbox is a 'scorecard' - a single clear plot with arrows/triangles showing whether the model version being tested is better or worse than a previous incarnation. Triangles pointing upward (green) indicate that the test model is better than the control and downward (purple) triangles indicate the control model is better. The area of the triangles is proportional to the absolute improvement (or deterioration) of the model and the triangles are outlined in black if the change is statistically significant at the 0.05 level determined using the Wilcoxon signed-rank test. The maximum triangle size, which occurs when the length of the base of the triangle is equal to the size of the square in which it is contained, is either set automatically (by selecting the maximum difference value from the data being compared) or can be done by manually setting a limit. The figures in this paper have the "max" values set automatically for each model comparison. The scorecards contain a huge amount of information, digested into an easy-to-understand summary. This allows fast assessments about model skill to be made, speeding up the evaluation (and therefore development) process. The model verification plotting comprises the FSS (score with spatial scale, score with forecast lead time, accumulation equivalent to particular centile with forecast lead time) and HiRA scores including bias at neighbourhood size with forecast lead time. Plotting of more traditional metrics (e.g. mean error and root-mean-square error (RMSE) at a grid point) was also included for a range of parameters (surface temperature, wind, relative humidity and 6-hourly precipitation amounts).

The diagnostic methods implemented in RMED toolbox version 1 also included domain (area) average plots (for a comprehensive set of meteorological diagnostics) especially useful for considering the diurnal cycle, histograms (for parameters such as screen temperature, wind, 3h mean rain rates and outgoing longwave radiation) for exploring distributions, and "cell statistics" (Hanley et al., 2015), a method for investigating the texture of a field through the application of a threshold to identify areas of exceedance or "cells". The number and size of the cells can then be analysed. This was first implemented to compare 3-hourly mean rain rates against GPM IMERG satellite data (Huffman, 2015, 2017), or if appropriate, UK radar data. The ability to create charts of model fields for a specific set of meteorological variables was also provided.

RAL1 provides a lot of detail due to its use at high resolution, but this can increase noise in traditional verification measures such as the root-mean-square error, which favours smooth fields over noisy ones. Multiple scores for the same parameter can be a source of confusion, providing different, even contradictory results. The RPS and FSS both evaluate hourly precipitation but they measure different attributes of the precipitation forecast. The FSS scores measures pattern, and the HiRA RPS focuses on intensity. It is possible to improve the forecast intensities whilst degrading the spatial pattern or texture of the forecast and this can lead to verification scores that are difficult to interpret. Murphy and Winkler (1987) stated the need for more than one independent score, measuring a range of forecast attributes to get a robust perspective of forecast performance.

## 5.2 Mid-Latitude performance over the UK

In this section we illustrate the impact of the RAL1 changes on model performance. The baseline used for the UK and mid-latitudes is RAL0. The UK evaluation consisted of a hierarchy of testing. Firstly, individual science changes (RMED tickets) were tested by running 100 case studies with a 1.5km horizontal grid-length, using the same domain as the Operational UKV model (Figure 3). These were simple downscaling runs (from the Met Office Global model) with no data assimilation. The cases sampled a wide range of meteorological conditions from the period July 2014 to April 2017 and comprised roughly equal numbers from each season. The cases were a mixture of poor forecasts (as identified by forecasters), high impact weather and normal everyday weather. The verification results from this stage of testing were used in the decision making process of whether individual science changes were performing well enough to progress to the next round of testing. Secondly, the tickets were packaged up into a "proto-RAL1" package and the same case study tests repeated. Typically there may be several "proto" packages trialled before a preferred package is chosen. Thirdly to test the impact of including data assimilation in RAL1, one month long UKV 3D-VAR Data Assimilation trials were run for Summer and Winter 2016. The exact choice of dates for the case studies (and indeed the data assimilation trials) can obviously affect the results, but the reason for running the case studies is to provide a relatively cheap and quick test of model changes before moving on to the more expensive data assimilation trials.

Figure 4 shows the HiRA scorecard comparing RAL1 performance with RAL0 for the 100 case studies and Figure 5 shows the results for the 3D-Var Winter and Summer trials. The first thing to note is that there is remarkably good agreement between the case study and the 3D-Var trial results. This shows that the case studies can give a good indication of likely performance in data assimilation trials and that the exact choice of dates is not crucial to the results provided enough cases are run. The second thing to note is that screen temperature is the variable that is (by far) the most significantly improved in RAL1.

Figure 6 shows the diurnal cycle of 1.5m temperature bias and RMSE for RAL1-M and RAL1-T against RAL0 for the 100 case studies. The figure shows that RAL1 reduces the bias and RMSE in the diurnal cycle of screen temperature. This addresses a long standing problem in the UKV model and is reflected in a statistically significant improvement to the Temperature RPS at most lead times in both case study (Figure 4, top row) and 3D-Var trials (Figure 5, top row). The improvement is primarily because of an increase in vegetation cover, at the expense of bare soil in RAL1, that reduces the thermal coupling between the atmosphere and soil. The reduction in scalar roughness lengths over grass tiles enhances the difference between skin and air temperatures. These changes lead to an amplified diurnal cycle of screen temperature and are supported by observational studies

at the Met Office Research Unit site at Cardington, near Bedford. The albedos of vegetated tiles are also reduced in RAL1 and this results in warmer daytime temperatures. These changes were all components of ticket 3 (see section 3.5). The impact on screen temperature varies according to the amount of vegetation present at a particular location. This is clearly illustrated by temperature differences over the UK shown in figure 7. In these plots the imprint of urban areas such as London show up as an area of little change between model versions RAL0 and RAL1. Another impact of the increase in vegetation cover from ticket 3 is that RAL1 reduces wind speeds (through an increase in the roughness length and therefore surface drag). The reduced wind speeds are beneficial at night time (reducing an overforecasting bias), but detrimental by day (Figure 8). Overall RAL1 shows statistically significant improvement to the 10m wind RPS at most lead times in both case study (Figure 4) and 3D-Var trials (Figure 5).

RAL1 gives an improvement to precipitation RPS at most lead times as seen in both case studies (Figure 4) and 3D-Var Summer trial (Figure 5 right panel). The 3D-Var Winter trial shows even stronger benefit with statistically significant improvements at all lead times (Figure 5 left panel). These HiRA results are based on raingauge data. 1hr FSS results (based on UK. radar as truth) for the case studies (Figure 10) show improvements to the 90th and 95th percentile results at all forecast ranges. The percentiles contain no bias information. However the absolute thresholds at 0.5mm, 1.0mm and 4.0mm in the hour generally show a detriment. The 6hr FSS for the case studies (Figure 11) show similar results and point to potentially undesirable changes to bias. The overall Precipitation Mean Error in the case studies is reduced in RAL1-M and this reduces an overforecasting bias (now shown). The 1mm frequency bias and 4mm frequency bias results (not shown) indicate that as we have reduced our mean error, we now on occasions have a frequency bias that is less than unity. RAL1 reduces the intensity of high precipitation rates (Figure 9) as a result of the moisture conservation change that removed spurious generation of precipitation by the semi-Lagrangian advection scheme from ticket 2, but this may have now revealed compensating errors.

RAL1 reduces the optical depth of fog as a result of the droplet taper change (ticket 1) and further discussion of fog processes and model performance can be found in Boutle et al. (2018). The case study results (Figure 4) and 3D-Var Summer trial (Figure 5 right panel) show an improvement to visibility RPS at all lead times except for T+3. The 3D-Var Winter trial shows even stronger benefit with statistically significant improvement at all forecast ranges (Figure 5 left panel). Figure 12 shows a fog case study with high pressure centred over N France. RAL1 has less extensive <100m fog over England where none is observed.

RAL1 reduces cloud amounts and raises cloud base. This is likely to be related to a drying of the boundary layer as a result of the moisture conservation change. Overall RAL1 shows statistically significant degradation to cloud fraction RPS at most lead times in both case study (Figure 4) and 3D-Var Winter trials (Figure 5 left panel). Subjective assessment of RAL1 by forecasters found that whilst largely very similar to RAL0, RAL1 tends to break up lower cloud faster than RAL0, especially where that cloud is fragmented. Whilst on average the reduction in cloud amounts verifies worse, in some cases it is good. Figure 13 shows a stratocumulus case from 23rd June 2015. RAL0 fails to break up the cloud cover through the daytime leading to excessive low/medium cloud. RAL1-M breaks up the cloud more accurately along with RAL1-T, with RAL1-T tending to have even less cloud than RAL1-M. RAL1-T uses the PC2 cloud scheme and this has been found to spuriously break up cloud in the UKV.

### 5.3 Tropical performance - Singapore

10 SINGV (Dipankar et al. in prep.) was a five year collaborative project between the Met Office and Meteorological Service Singapore, which ran from 2013 to 2018. For the duration of the project the SINGV domain was the focal point for convective-scale model development in the tropics, and it was within this framework that the differences between RAL1-T and RAL1-M were identified, tested and then implemented. In this section we illustrate the impact of the changes implemented over the course of the SINGV project by comparing the performance over Singapore of the RAL1-T and RAL1-M configurations.

15 The model development trialling strategy within SINGV focused on downscaling global model forecasts, i.e. using the case study approach described above for UK testing. In order to reduce the potential dependency of the results on the choice of case, a whole month of forecasts were run out to T+36 initialised from every 00z and 12z analysis. This approach ensured that summary measures were as robust as possible, whilst individual forecasts could be assessed in detail.

Figure 14 shows results for November 2016. Three model configurations are shown:- (i) RAL1-T, (ii) RAL1-T-mPC2, 20 which is RAL1-T, but using the RAL1-M cloud scheme, and (iii) RAL1-T-3xBL, which is RAL1-T but with the RAL1-M boundary layer settings. With these configurations we are able to illustrate the impact of the key differences between RAL1-T and RAL1-M. In Figure 14(a) it is evident that the peak in the diurnal cycle of rainfall is too early compared to GPM for all three configurations. However, the time of convective initiation (when the rainfall first begins to increase, i.e. at T+15) is well captured by RAL1-T and RAL1-T-mPC2. In contrast RAL1-T-3xBL initiates even earlier (approximately two hours) than 25 RAL1-T. Other experiments (not shown) indicate that both the activation of the stochastic perturbations and the change to the convective BL stability functions contribute to this degradation in performance.

Figure 14(b) shows the impact on the rainfall FSS of removing PC2 from the RAL1-T configuration. The impact is large and shows that switching from the PC2 cloud scheme and reverting back to the Smith cloud scheme significantly reduces the ability of the model to skilfully predict high-impact rainfall events. Figure 14(c) shows that the impact of the BL differences is 30 also to reduce the skill of the model and this signal is significant for the high percentile threshold for the majority of lead times.

An illustration of the differences in the RAL1-T and RAL1-M rainfall distribution over Singapore is shown in Figure 15, which shows snapshots of the model forecasts for a single case study for 18th August 2016 compared to the Changi radar. The rainfall maps for early afternoon local time shown in Figure 15(a)-(c) further illustrate the benefit of deactivating the stochastic perturbations in RAL1-T. The RAL1-T rainfall map compares favourably with the radar estimated rainfall hourly accumulation with, in both cases, isolated convection just starting to develop over the Malay Peninsula. In contrast, the RAL1-M rainfall map shows spurious localised convection has been initiated over a large area. This spurious convection has been triggered by the combined effect of the stochastic perturbations and the change to the convective boundary layer stability functions (as confirmed by additional experiments, not shown).

5 The rainfall maps for early the following morning local time (Figure 15(d)-(f)) show a Sumatran Squall passing through Singapore. The improved location of the squall in the RAL1-T forecast is typical of the impact found when the PC2 cloud scheme is implemented in SINGV. The impact of PC2 is to increase light rain amounts and decrease very heavy rain amounts compared to the Smith scheme. Effectively this makes the model more dissipative and this leads to a reduction of small-scale

structure which enables the large-scale envelope of features like Sumatran Squalls to be better handled and hence to propagate
more realistically. The increased free-atmospheric mixing further increases the dissipation and the two together were found to improve the ability of the model to propagate Sumatran Squalls faster and further, rather than have them not develop or dissipate prematurely.

## 5.4 Tropical performance - Darwin MCS case

The Australian evaluation was carried out by the Bureau of Meteorology in Australia and consisted of running 8 case studies over various domains with a 1.5km horizontal grid-length. Here, we discuss one of the 8 cases and compare both RAL1-T and RAL1-M against radar observations.

The observations come from the Darwin C-band polarimetric radar which collects 3D observations out to a range of 150 km (Louf et al., 2018), which allows for a detailed evaluation of simulated tropical convection. (Figure 16 shows the domain the radar covers and the area over which the comparison with the model is done). The case studied is 18 February 2014 where active monsoon conditions produced a mesoscale convective system (MCS). The monsoon trough was stalled at the base of the Top End (geographical region encompassing the northernmost section of the Australia's Northern Territory), and there was a deep moisture layer and low-level convergence. The observed and modelled MCS lifecycle is illustrated in the timeries plots of Figure 17, which shows the fractional area of the radar domain covered by reflectivities greater than 10 dBZ as a function of height and time over a 12-hour period. From 12 - 15 UTC scattered convection was observed around Darwin and the observed spatial coverage of cloud and rain within the radar domain increased from 20 to 40%. By 17 UTC the convection had become organised with numerous cells and a cloud shield exceeding 200 km in diameter. At 18UTC the deepest convection was observed, with 10 dBZ cloud top heights around 13 km. After this time, the mostly oceanic MCS matured and was composed of an extensive stratiform cloud region. The 1.5 km horizontal grid-length simulations using RAL1-M and RAL1-T show deeper clouds and more extensive cloud and rain area coverage at earlier times than the radar observations. The cloud top heights peak at 15 UTC in the RAL1-M simulation at a height greater than 14 km, which is 3 hours earlier than the observed cloud top height maximum and about 1.5 hours earlier than the RAL1-T simulation. RAL1-M fails to produce significant fractional areas of cloud and rain greater than 0.8 throughout the MCS lifecycle, whereas RAL1-T shows a better representation of extensive stratiform cloud and rain areas, albeit a couple of hours too early. Both simulations overestimate the rainfall at the surface across the radar domain (Figure 16), which is due to too many areas of heavy rain > 8 mm/hr (not shown). The timing of the observed domain mean rainfall maximum occurs about an hour after the deepest clouds and a couple of hours before the hydrometeor spatial cover is maximal. While the simulations capture the same sequence of events, the rate of change in domain mean rain rate as the system evolves from a developing to a mature MCS is amplified. This is primarily due to the model overestimating rainfall during the developing stages that are dominated by deep convection. RAL1-T produces a larger overestimate in total precipitation than RAL1-M but more accurately represents the timing of the MCS lifecycle of precipitation.

## 5.5 Tropical performance - Tropical cyclones in the Western North Pacific

Evaluation of RAL1 for tropical cyclone (TC) forecasting concentrated on the Philippines since this is the most exposed country in the world to TCs. Figure 18 shows the regional model domain used. This has a large extent to the east of the Philippines to ensure that TCs travelling northwest towards the islands are captured in the domain long before making landfall.

A total of 130 TC forecasts (initialisation times between 15 March and 16 December 2015) were produced with each of RAL1-T and RAL1-M, using the domain shown in Figure 18 with a horizontal grid-length of 4.4 km and the $L80(59_t, 21_s)_{38.5}$ vertical level set. Storms were tracked in model output using the Met Office TC tracker (Heming, 2017) and only storm cases appearing in both experiments were kept to ensure a fair comparison. A number of cases had two storms present in the domain at T+0.

Figure 19 shows the mean bias (model - obs) in TC maximum surface wind speed and central pressure as a function of forecast lead time for the two RAL1 models. It is clear that both configurations give very similar intensity predictions. There is a protracted spin-up period as the regional models adjust from the weak initial state inherited from the driving global model. During this time, intensity errors steadily reduce and, beyond $T + 36$, the bias in wind speed is close to zero (although this is the result of compensating errors: surface winds are typically under-estimated in storms of category 3 and above, but over-estimated in weaker storms). However, RAL1 has a tendency to over-deepen storms, with central pressures dropping below those observed at about $T + 24$, asymptoting to a value approximately $10 - 15$ hPa too low beyond $T + 48$. This could be due, at least in part, to the lack of ocean feedback on the atmosphere in the model. The differences in mean intensity biases visible beyond $T + 72$ are not statistically significant owing to the declining sample size with lead time.

It follows from Figure 19 that the dynamical relationship between the wind and pressure fields in the model must be different to that observed. To highlight this, Figure 20 shows scatterplots of maximum surface wind speed and central pressure for the RAL1 configurations, along with the observed wind-pressure relation (WPR) derived from Joint Typhoon Warning Center (JTWC) best-track data.

The RAL1 relations are a good match to the observed WPR up to wind speeds $\sim 100$ knots, but are too steep beyond this. In other words, wind speeds in strong storms are too slow for their central pressure. This is likely because air-sea drag is currently over-estimated in the model at high wind speeds. Plans for RAL2 include a reduction of the drag coefficient at high wind speeds, consistent with available observations.

Figure 21 displays the mean error in storm position relative to observations (as measured by the direct positional error, DPE) as a function of forecast lead time for the RAL1 models. Track errors in RAL1-T and RAL1-M are broadly comparable. In both cases the DPE increases by approximately 36 km per day of forecast, reaching a maximum of around 200 km at $T + 120$. There is a hint that RAL1-T may give more accurate track predictions, but the current sample is too small for this to be a statistically significant result.

## 5.6 Regional model ensemble performance for USA Hazardous Weather Testbed

The Met Office has been involved in the US Hazardous Weather Testbed (HWT) Spring Forecasting Experiment (Kain et al., 2017), held annually in Norman, Oklahoma, for a number of years. UM kilometre grid scale regional models have been run and their performance has been found to be very competitive with the locally developed models (Kain et al., 2017). The meteorology of the midwest U.S.A with its severe weather (tornadoes, hail etc.) is different from that of the mid-latitudes (section 5.2) and the tropics (section 5.4). This is a good test for the regional model and ensures that we don't tune the model for a narrow set of meteorology. In addition the expertise of the HWT Forecast team and the excellent observational network allow a robust assessment of model performance.

After the 2017 HWT, a 12-member 2.2km grid-length UM ensemble was generated by one-way nesting the US 2.2 km domain (run routinely for the HWT) within the 12-member global ensemble (MOGREPS-G). The case studied was 16 May 2017, with MOGREPS-G initialised at 0000 UTC on this day. MOGREPS-G had initial condition perturbations and used the Random Parameters (RP) scheme (McCabe et al., 2016) to perturb the model physics. Initial conditions and LBCs for each 2.2km ensemble member were obtained from the corresponding global member. The RP scheme was not used in the 2.2km ensemble, members were purely downscaled from the global. Each global ensemble member drove two 2.2km ensemble members, each with a different science configuration: RAL1-M and RAL1-T. On this day there was a trough situated over the southern Rockies which was moving eastward, with a converging dryline across the Midwest and a strengthening low level jet. Convection initiated over Texas at around 1800 Z (1300 CDT) and upscaled very quickly with supercells observed over Oklahoma. Figure 22 shows the hourly-accumulated precipitation averaged over the Texas-Oklahoma region for 16-17 May 2017 for the RAL1-M and RAL1-T ensembles respectively. These figures highlight differences in the convection initiation time between the two configurations. Compared with the radar observations the RAL1-M members tended to initiate too early and produced a peak in precipitation at around 2000-2100 UTC that was not observed by the radar. Conversely the RAL1-T ensemble members tended to initiate too late. Switching off the stochastic perturbations in the RAL1-M ensemble resulted in about a 1-hour delay in the onset of precipitation and reduced the precipitation peak (not shown). However, the onset of precipitation was still not as delayed as it was in the RAL1-T ensemble, suggesting the mixing length differences also contribute to the initiation time differences between RAL1-M and -T. Overall, the RAL1-T ensemble seems to better capture the supercells on this day, with more members simulating supercell-like features (Hanley and Lean in prep.).

## 6 Conclusions

The definition of RAL1 is an important step in the development of kilometre grid scale configurations of the Unified Model. By concentrating the model development effort on a well-defined system, the model users are better placed to learn from each other and to identify and resource the main priorities for future model development. In this paper we have defined configurations of the regional Met Office Unified model, described a "toolbox" that allows us to evaluate its performance and provided some baseline tests to give a bench mark of performance. Performance is tested in both simulations with Data Assimilation and without - the latter we refer to as case studies.

While it remains an ambition to have a single configuration of the model that works across all regions, at this stage we have defined two: RAL1-M for mid-latitudes and RAL1-T for the tropics. Both are clearly documented in terms of the model physics and their performance in relevant regions. For the mid-latitude system the most recent developments are described in more detail and the NWP performance changes due to these recent changes are shown. To do this we have defined a previous operational NWP version of the Unified Model which we refer to here as RAL0. The performance of the tropical system is presented as a benchmark for future developments.

The recent science developments included in RAL1-M are shown to significantly improve two long-standing issues with model performance in NWP. The inclusion of moisture conservation reduces overly intense local precipitation rates and the changes to land use and vegetation properties improve a damped diurnal cycle in near-surface temperatures. We also see modest improvements to forecasts of low visibilities. The conservation of moisture was of particular importance to the tropical configuration of the model although this was not shown in the paper.

A goal of having a clearly defined version of a regional model, and perhaps more importantly a series of tests for that model that gives confidence that changes are generally improving the system is hugely challenging. In this paper we have shown a series of tests in a small number of regions that requires substantial computational effort. Yet, we have only sampled a small fraction of the types of meteorology that the model should be expected to represent. Looking ahead, we need to consider other regions such as the poles and more broadly sampling the range of weather types seen in the regions we have considered. One very specific area which is not covered in this paper is the performance of the model in climate simulations. It remains a high priority to include climate testing in the development process of the regional model although with the high computing costs involved in regional climate runs at the kilometre gridscale system the test will need careful design.

Looking ahead, in addition to improving the modelling system, consolidating regional differences and documenting this we also aim to substantially improve the evaluation process. This will include climate testing, increased used of ensembles and the testing in more regions. This will require concerted effort and coordination from the partnership developing the RAL configuration, but this should lead to a better understanding of its strengths and weaknesses, and lead to the more efficient development of further improvements.

## 7   Code availability

Due to intellectual property right restrictions, we cannot provide either the source code or documentation papers for the UM or JULES.

*Obtaining the UM.* The Met Office Unified Model is available for use under licence. A number of research organisations and national meteorological services use the UM in collaboration with the Met Office to undertake basic atmospheric process research, produce forecasts, develop the UM code and build and evaluate Earth system models. For further information on how to apply for a licence see http://www.metoffice.gov.uk/research/modelling-systems/unified-model.

*Obtaining JULES.* JULES is available under licence free of charge. For further information on how to gain permission to use JULES for research purposes see http://jules-lsm.github.io/access_req/JULES_access.html.

*Details of the simulations performed.* UM/JULES simulations are compiled and run in suites developed using the Rose suite engine (http://metomi.github.io/rose/doc/rose.html) and scheduled using the cylc workflow engine (https://cylc.github.io/cylc/). Both Rose and cylc are available under v3 of the GNU General Public License (GPL). In this framework, the suite contains the information required to extract and build the code as well as configure and run the simulations. Each suite is labelled with a unique identifier and is held in the same revision controlled repository service in which we hold and develop the model's code. This means that these suites are available to any licensed user of both the UM and JULES. We document a set of reference RAL1-based simulations in Table 3.

**Table 3.** Identifiers for a set of RAL1 reference simulations across a number of systems/applications. These suites are held on the Met Office Science Repository Service, which also holds the UM and JULES code. %

| Application | Suite id | UM version/JULES version |
|---|---|---|
| UK case studies | u-ao109 | UM10.6/JULES4.7 |
| Singapore case studies | u-av356 | UM10.9/JULES5.0 |
| Darwin MCS case study | u-ax904 | UM10.6/JULES4.7 |
| Tropical cyclone case studies | u-aq686 | UM10.6/JULES4.7 |
| USA Hazardous Weather Testbed case study | u-ao861 | UM10.6/JULES4.7 |

## 8 Appendix

We document a list of acronyms in Table 5.

We document the list of RMED tickets included in RAL1 in Table 6.

*Competing interests.* The authors declare that they have no conflict of interest.

*Author contributions.*

Mike Bush led the RAL1 testing and evaluation process and prepared the manuscript with contributions from all co-authors. Tom Allen, Ian Boutle, John Edwards, Adrian Lock, James Manners, Cyril Morcrette, Jonathan Wilkinson, Nigel Wood and Mohamed Zerroukat are either code owners and/or developers of the model code included in RAL1.

Caroline Bain, Anke Finnenkoetter, Charmaine Franklin, Kirsty Hanley, Humphrey Lean, Marion Mittermaier, Rachel North, Chris Short, Stuart Webster and Mark Weeks performed the evaluation. Jon Petch, Simon Vosper and David Walters contributed to the writing of the Introduction and Conclusions sections.

---

**Algorithm 1** Iterative structure of time step $(n+1)$. Here, we use two inner and two outer loops ($L = 2$, $M = 2$).

---

1: Given the model's state at the previous time step $n$, and let the first guess of the prognostic variable $F^{n+1}$ at time level $(n+1)$ be

$$F^{n+1} = F^n$$

2: For LAMs the appropriate LBCs are also provided for time step $(n+1)$ (note that LBCs are interpolated in time and space and hydrostatically balanced)

3: If LAM, fill external halos with the appropriate data [external halos contain constant values extended outward from adjacent LBC data (e.g., zero gradient assumption, see text)]

4: Compute slow parametrised processes and time level $n$ forcings $R_F^n$

5: **for** $m = 1, M$ **do** {*departure (outer-loop) iteration*}

6:     For LAM, update dynamics (or wind) only LBCs to avoid the back-substitution wind diverging from LBCs (see text)

7:     Compute departure points using wind at time level $n$ and the latest estimate at time level $(n+1)$

8:     SL advect all prognostic variables (requires interpolation to departure points)

9:     SL advect moisture fields

10:     Apply ZLF to restore moisture mass conservation after advection (if required)

11:     If ($m = M$), SL advect all other tracers and apply ZLF (if required)

12:     Compute fast parametrised processes using the latest $F^{n+1}$

13:     Evaluate all Helmholtz terms invloving time level $n$ (this includes terms interpolated at departure points)

14:     **for** $l = 1, L$ **do** {*non-linear (inner-loop) iteration*}

15:         Evaluate non-linear terms involving the latest state $(n+1)$

16:         Evaluate time level $(n+1)$ components of Helmholtz right hand side $RHS^{n+1}$

17:         For LAM, adjust $RHS^{n+1}$ to satisfy the LBCs pressure values (note also that the Helmholtz coefficients are also modified to enforce a Dirichlet type boundary condition (known pressure at the LBC at time level $(n+1)$))

18:         Solve the Helmholtz problem for the pressure increment $\pi'$ and evaluate $\pi^{n+1} \equiv \pi^n + \pi'$

19:         Compute the other prognostic variables at time level $(n+1)$ via the back-substitution

20:     **end for**

21: **end for**

22: For LAM, update all LBCs (i.e., blend LBCs data and model's solution)

---

**Table 4.** Source datasets used to create standard ancillary files used in RAL0. [%]

| Ancillary field | Source data | Notes |
| --- | --- | --- |
| Land Sea mask | IGBP; Loveland et al. (2000) | Used for UKV/MOGREPS-UK. |
| | CCI; Hartley et al. (2017) | CCI mask lacking in inland lakes definition |
| Mean/sub-grid orography | DTED 1km ; | Used for UKV/MOGREPS-UK. |
| | GLOBE 30″; Hastings et al. (1999) | Fields filtered before use |
| | SRTM; Bunce et al. (1996) | Shuttle Radar Topography Mission. Mean orography only. |
| | | Available up to 60 degrees North. |
| Land usage | IGBP; Loveland et al. (2000) | Mapped to 9 tile types |
| | ITE; Bunce et al. (1996) | U.K. only |
| | CCI; Hartley et al. (2017) | European Space Agency Land Cover Climate Change Initiative |
| Soil properties | HWSD; Nachtergaele et al. (2008) | Three datasets blended via optimal interpolation |
| | STATSGO; Miller and White (1998) | |
| | ISRIC-WISE; Batjes (2009) | |
| Leaf area index | MODIS collection 5 | 4 km data (Samanta et al., 2012) mapped to 5 plant types |
| Plant canopy height | IGBP; Loveland et al. (2000) | Derived from land usage and mapped to 5 plant types |
| Bare soil albedo | MODIS; Houldcroft et al. (2008) | |
| SST/sea ice | System/experiment dependent | |
| Ozone | Li and Shine (1995) | |
| Murk aerosol | NAEI, ENTEC and EMEP emission inventories | |
| CLASSIC aerosol climatologies | System/experiment dependent | Used when prognostic fields not available |

*Acknowledgements.* The development and assessment of the Regional Atmosphere Land configurations is possible only through the hard work of a large number of people that exceeds the list of authors. Specifically we would like to thank Belinda Roux and Hongyan Zhu [2], Magdalena Gruziel-Slomka and Malgorzata Melonek [3], Seungwoo Lee [4], Jayakumar [6], Stuart Mooore [7] and Estelle Marx and Elelwani Phaduli [8] for running case studies and evaluating model performance. Douglas Boyd [1], George Pankiewicz [1], Charmaine Franklin [2], Magdalena Gruziel-Slomka [3], Seungwoo Lee [4], Jeff Lo [5], Saji Mohandas [6], Stuart Moore and Trevor Carey-Smith [7], Stephanie Landman [8] and Steven Rugg [9] for the coordination, led by Mike Bush [1], of RAL1 testing and evaluation around the UM Partnership.

[1] Met Office, Exeter, UK

[2] Bureau of Meteorology (BoM), Melbourne, Victoria, Australia

[3] Interdisciplinary Centre for Mathematical and Computational Modelling (ICM), Warsaw, Poland

[4] Korea Meteorological Administration (KMA), Seoul, South Korea

[5] Meteorological Service Singapore (MSS), Singapore

[6] National Centre for Medium Range Weather Forecasting (NCMRWF), Noida, India

[7] National Institute of Water and Atmospheric Research (NIWA), Wellington, New Zealand

**Table 5.** Acronym list. %

| Acronym | Meaning | Notes |
| --- | --- | --- |
| EACF | Empirically Adjusted Cloud Fraction | |
| ENDGame | Even Newer Dynamics for General atmospheric modelling of the environment | Dynamical core used in RAL0 and RAL1 |
| GA | Global Atmosphere | Global Atmosphere science configuration |
| GA3.1 | Global Atmosphere 3.1 | A specific GA science configuration |
| GA7.0 | Global Atmosphere 7.0 | A specific GA science configuration |
| GL | Global Land | Global Land science configuration |
| GL7.0 | Global Land 7.0 | A specific GL science configuration |
| JULES | Joint UK Land Environment Simulator | Community Land surface model |
| LAM | Limited Area Model | |
| LBCs | Lateral Boundary Conditions | |
| LW | Longwave | |
| MOGREPS-UK | Met Office Global and Regional Ensemble system - UK | UK NWP operational ensemble system |
| NMS | National Met Services | |
| NWP | Numerical Weather Prediction | |
| RAL | Regional Atmosphere and Land | |
| RAL0 | Regional Atmosphere and Land 0 | Baseline RAL science configuration |
| RAL1 | Regional Atmosphere and Land 1 | First RAL science configuration |
| RAL1-M | Regional Atmosphere and Land 1 - Mid Latitudes | |
| RAL1-T | Regional Atmosphere and Land 1 - Tropics | |
| RMED | Regional Model Evaluation and Development | |
| SI | Semi-implicit | |
| SL | Semi-Lagrangian | |
| SOCRATES | Suite Of Community RAdiative Transfer codes based on Edwards and Slingo | Radiative Transfer scheme |
| SW | Shortwave | |
| UKV | UK Variable (resolution) | UK NWP operational deterministic model |
| UM | Unified Model | |
| ZLF | Zero Lateral Flux | |

[8] South Afican Weather Service (SAWS), Pretoria, South Africa

[9] United States Air Force (USAF), 557th Weather Wing, Offutt Air Force Base, Nebraska, United States of America

The research/project work of Charmaine Franklin was undertaken with the assistance of resources and services from the National Computational Infrastructure (NCI), which is supported by the Australian Government.

The GPM IMERG Late Precipitation L3 Half Hourly 0.1 degree x 0.1 degree V04 data were provided by the NASA/Goddard Space Flight

**Table 6.** RMED tickets included in RAL1. %

| RMED Ticket number | RAL1-M/RAL1-T | Description of RAL1 change |
| --- | --- | --- |
| 1 | RAL1-M and RAL1-T | Improved droplet number profile in the lower boundary layer |
| 2 | RAL1-M and RAL1-T | Conservative advection for moist prognostics |
| 3 | RAL1-M and RAL1-T | Improvements to land usage and vegetation properties |
| 5 | RAL1-M and RAL1-T | Improved representation of mixing across the boundary layer top |
| 9 | RAL1-M and RAL1-T | Improved treatment of gaseous absorption |
| 10 | RAL1-M and RAL1-T | Reductions in sensitivity to vertical resolution |
| 11 | RAL1-M and RAL1-T | Change method of PMSL calculation to be more efficient |
| 12 | RAL1-M | Revision of free-atmospheric mixing length |
| 15 | RAL1-M and RAL1-T | Retuning BL mixing across the LCL in cumulus |
| 16 | RAL1-T | Cloud Scheme Upgrades |
| 19 | RAL1-M and RAL1-T | Correct inappropriate treatment of graupel as fresh snow in JULES |
| 25 | RAL1-M | Updates to stochastic boundary layer perturbations |
| 26 | RAL1-T | Revised unstable stability functions |

Center's Goddard Earth Sciences Data and Information Services Center and PPS, which develop and compute the GPM IMERG Late Precipitation L3 Half Hourly 0.1 degree x 0.1 degree as a contribution to GPM, and archived at the NASA GES DISC.

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

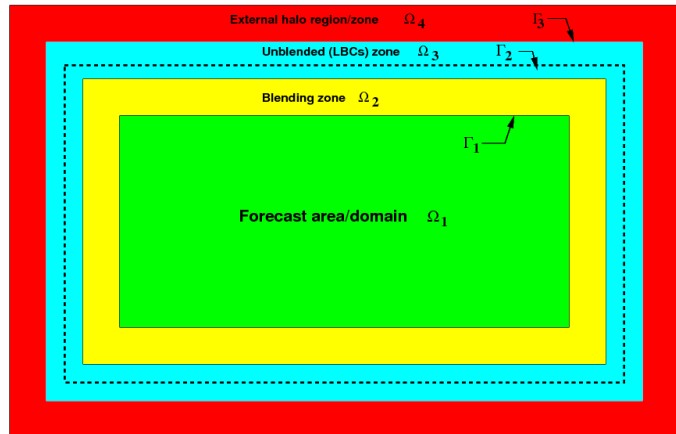

**Figure 1.** Schematic of the LAM configuration. In this configuration a LAM with a physical (or forecasting) region denoted by $\Omega_1$ is shown in green. On the periphery of the forecasting area there is an extended computational domain ($\Omega_E = \Omega_2 + \Omega_3 + \Omega_4$) that includes a blending (yellow) zone $\Omega_2$, an unblended (blue) zone $\Omega_3$ and an external halo (red) zone $\Omega_4$ (which arise from the parallel domain decomposition). Note that in general the relative sizes of ($\Omega_2, \Omega_3, \Omega_4$) are a lot smaller than $\Omega_1$, but they are exaggerated here for clarity. Also the use of the word RIM refers to the whole size of LBCs which are all the grid-points that lie in the region $\Omega_R = \Omega_2 + \Omega_3$ (yellow and blue).

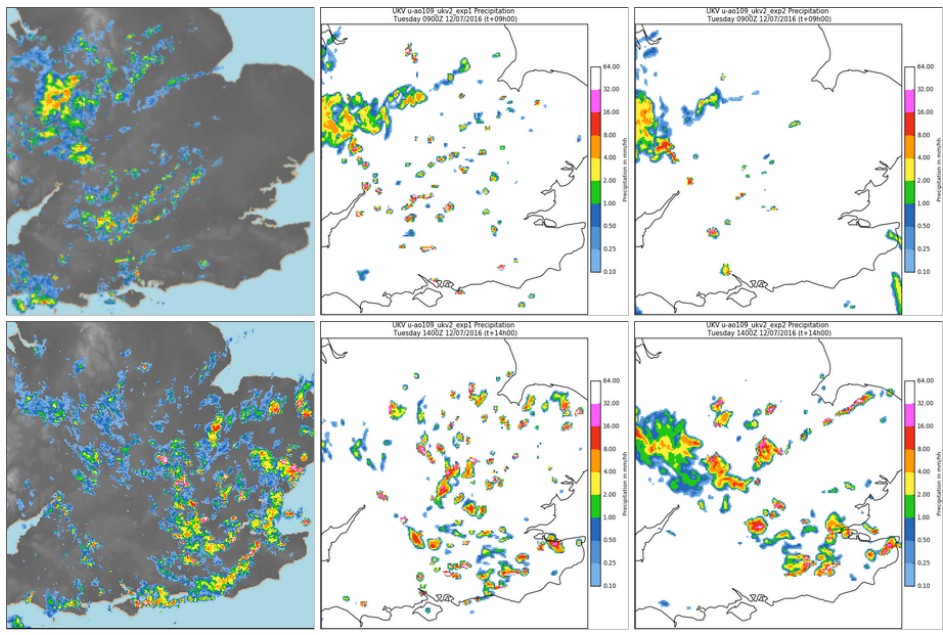

**Figure 2.** 12th July 2016 case study showing radar (left), RAL1-M (middle)and T (right) at 09Z (top) and 14Z (bottom) for a case of showers in the UK.

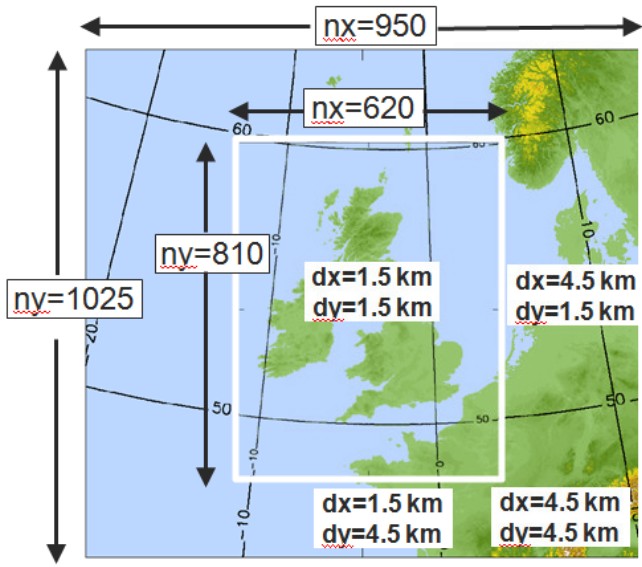

**Figure 3.** Domain for UK Case studies.

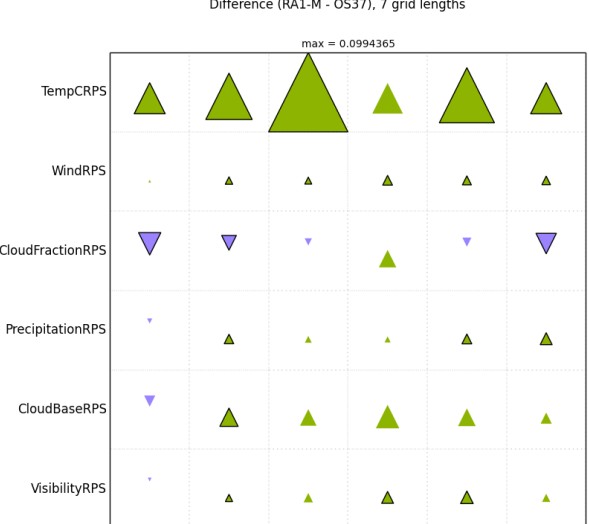

**Figure 4.** Case studies: RAL1-M vs RAL0 HiRA summary scorecard at 10.5km (7 grid-lengths) spatial scale. HiRA uses synoptic observations (see section 5).

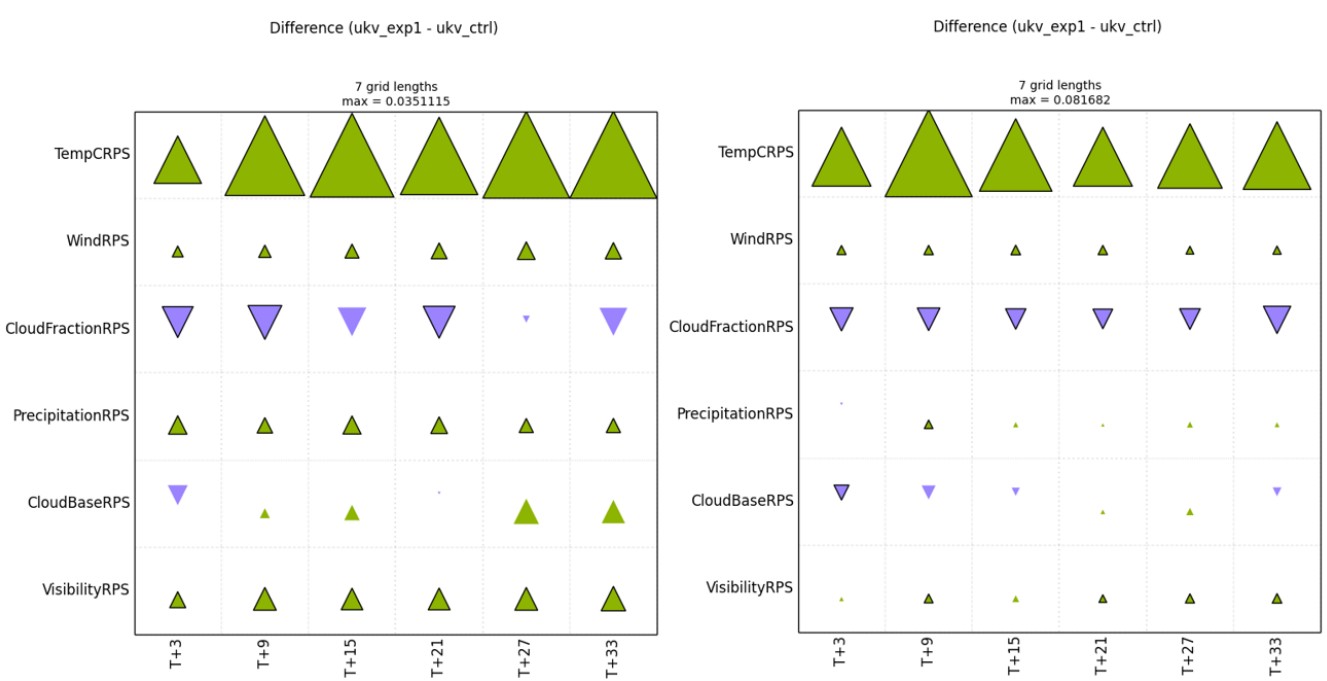

**Figure 5.** 3DVAR trials: RAL1-M vs RAL0 HiRA summary scorecard at 10.5km (7 grid-lengths) spatial scale for Winter (left) and Summer (right). HiRA uses synoptic observations (see section 5).

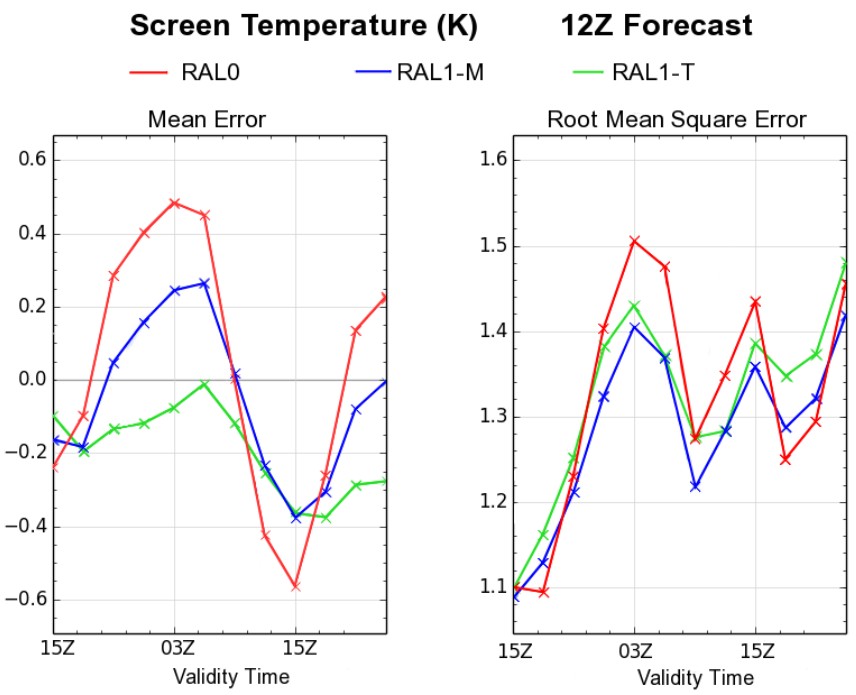

**Figure 6.** Case studies: Diurnal cycle of Screen Temperature bias (left panel) and RMSE (right panel) for 12Z forecasts. RAL0 (red), RAL1-M (blue) and RAL1-T (green).

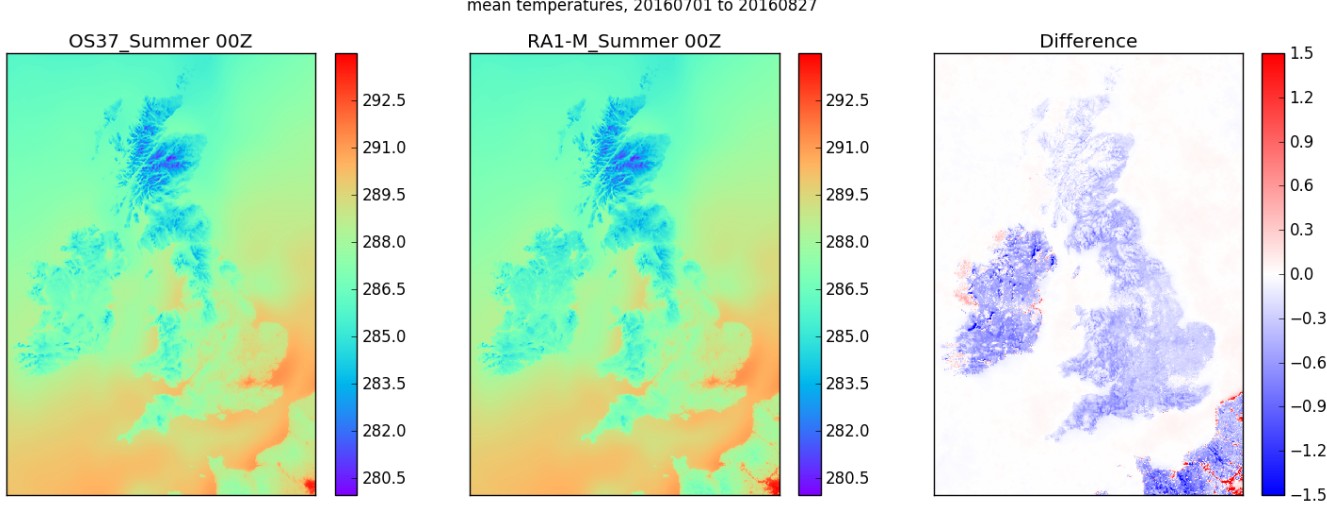

**Figure 7.** UKV Model: Summer Mean temperature differences at 00Z for RAL0 (left panel), RAL1 (middle panel) and RAL1-M minus RAL0 (right panel).

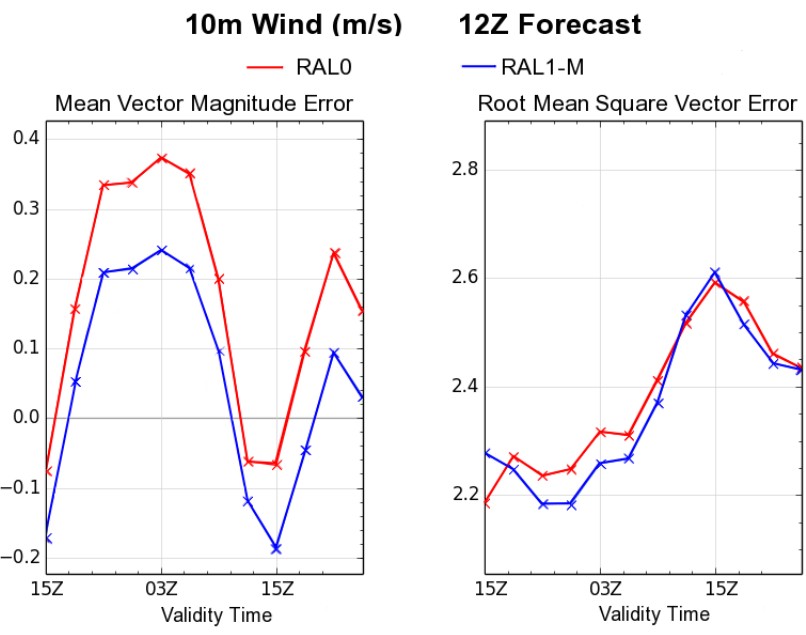

**Figure 8.** Case studies: Diurnal cycle of 10m wind Mean vector magnitude error (left panel) and Root Mean Square Vector Error (right panel) for 12Z forecasts. RAL0 (red) and RAL1-M (blue).

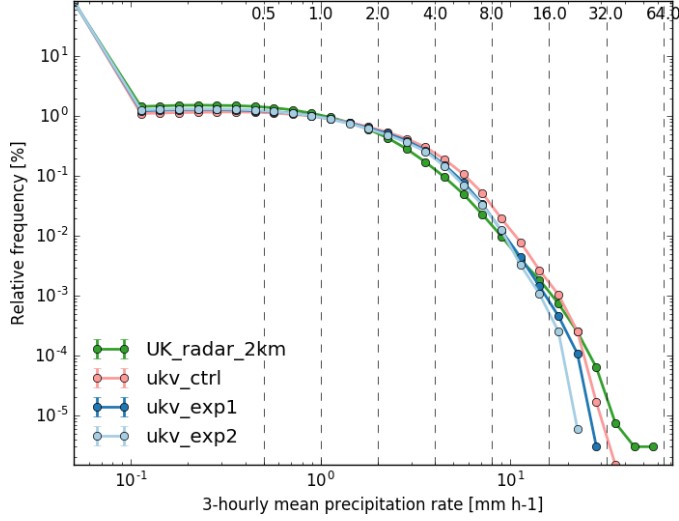

**Figure 9.** Case studies: Relative frequency of 3-hourly precipitation rate. RAL0 (red), RAL1-M (dark blue), RAL1-T (light blue) and 2km U.K radar (dark green).

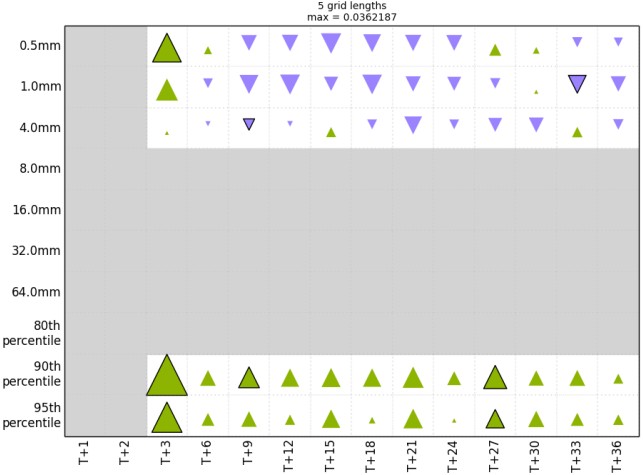

**Figure 10.** FSS at 5 grid-lengths for 1 hour accumulations.

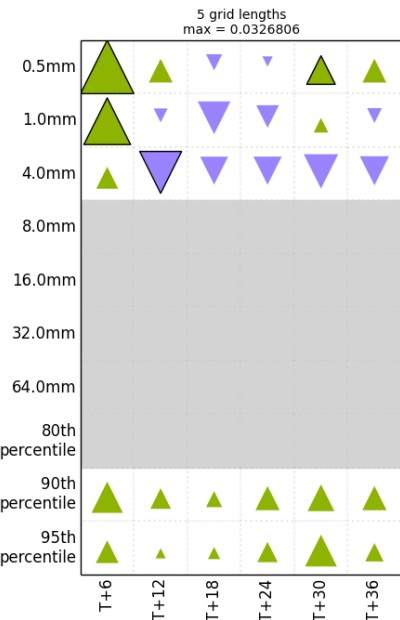

**Figure 11.** FSS at 5 grid-lengths for 6 hour accumulations.

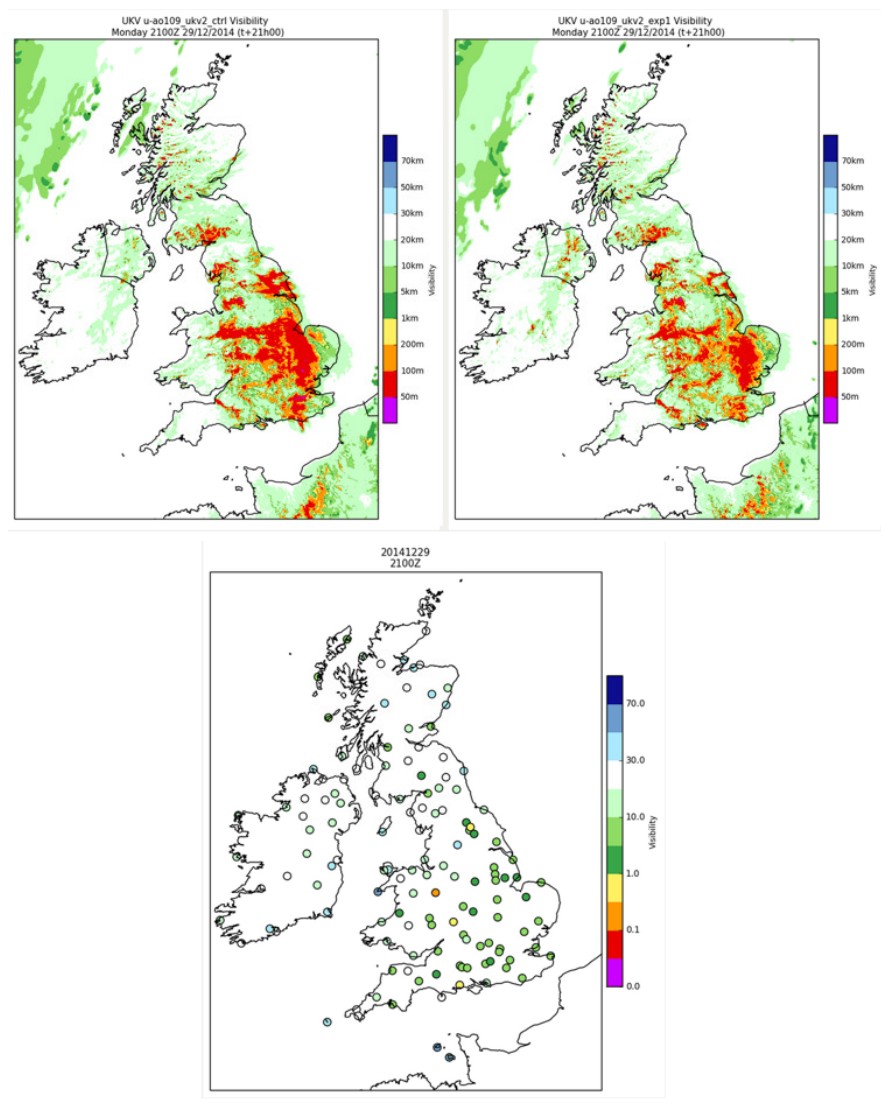

**Figure 12.** Fog case study 29th December 2014: Visibility from RAL0 (left) and RAL1-M (right) at T+21 VT 21Z on 29/12/14 and corresponding station obs.

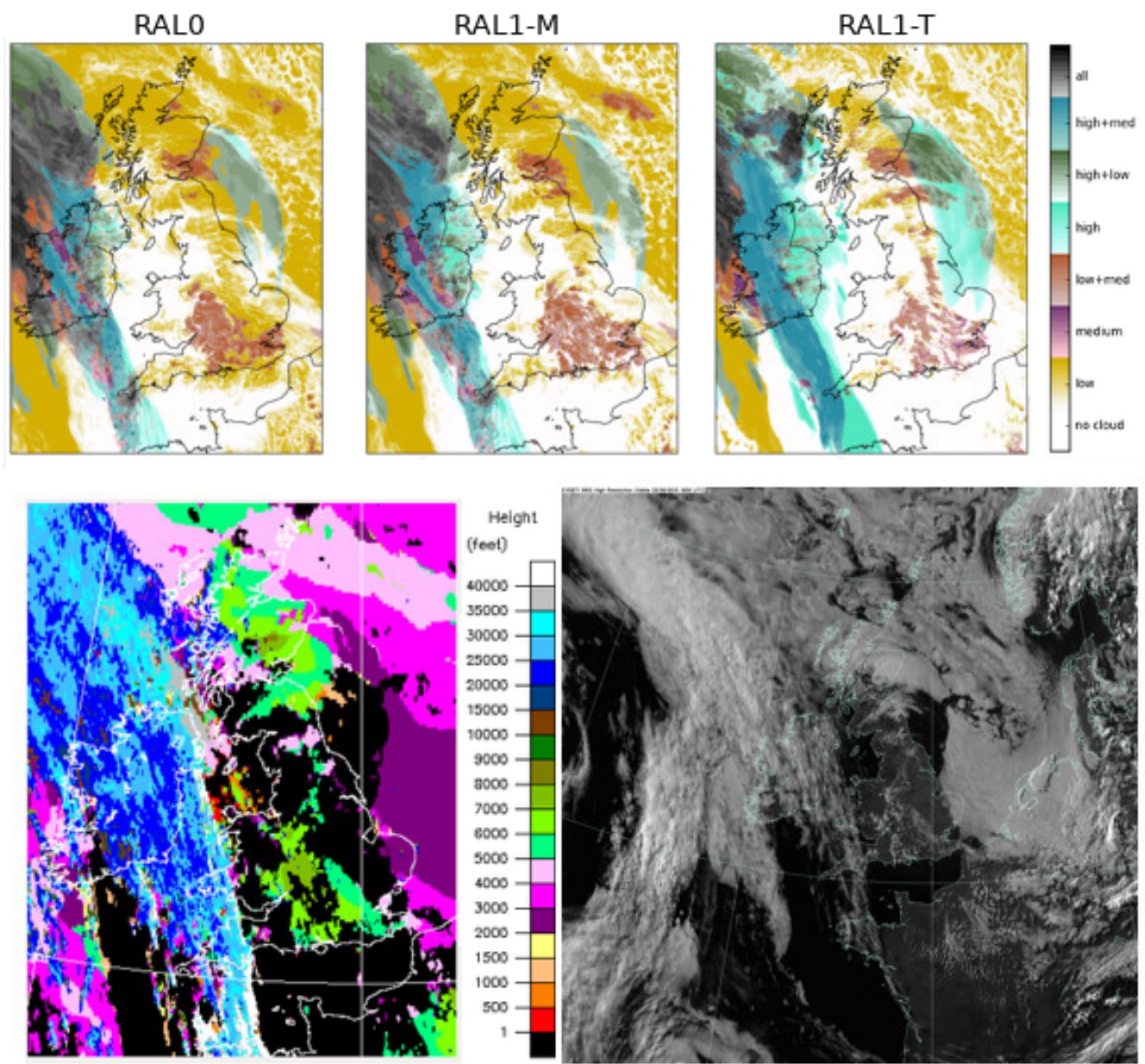

**Figure 13.** Cloud cover at T+18 (stratocumulus case) showing RAL0 (left), RAL1-M (middle) and RAL1-T (right) on 18z 23/06/15 with corresponding satellite imagery (false colour and visible).

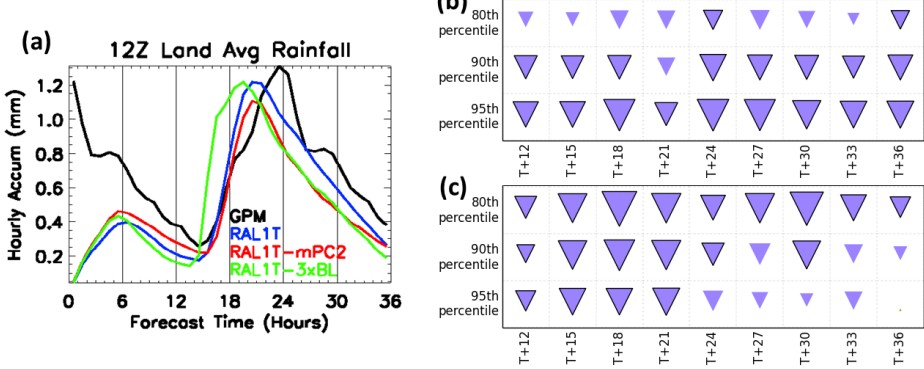

**Figure 14.** (a) Domain averaged rainfall timeseries for all forecasts initialised at 12z for November 2016. (b) Hinton plot showing the difference in FSS due to removing ticket 16 (RAL1-T-mPC2 minus RAL1-T) for a 250 km length-scale compared to 3 hourly GPM. (c), as (b), but showing the impact on FSS of reverting the three BL changes back to the RAL1-M settings (RAL1-T-3xBL minus RAL1-T). Note that on the Hinton plots downward pointing purple triangles indicate that RAL1-T is more skilful, whilst solid lines around the triangles denote that the difference is statistically significant.

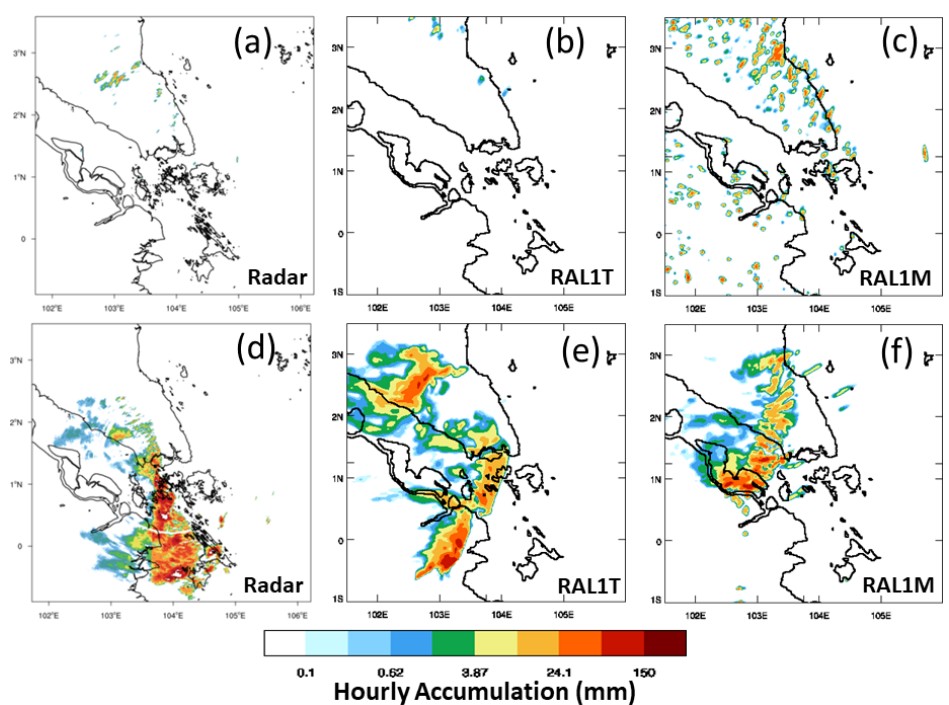

**Figure 15.** (a) Hourly rainfall accumulation for the Singapore radar for 06z 18/08/2016. (b) as (a), but showing the T+18 RAL1-T forecast initialised at 12z 17/08/2016 (so valid at the same time as the radar). (c), as (b), but showing the equivalent RAL1-M imagery. (d)-(f), as (a)-(c), but for the hourly accumulation for 00z 19/08/2016, which is T+36 for the same model forecasts as shown in (b) and (c).

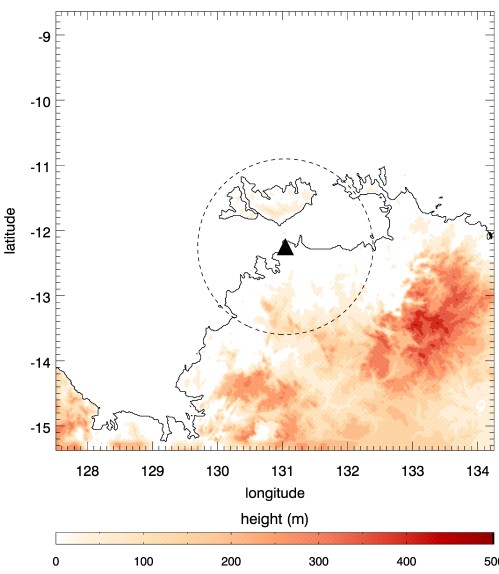

**Figure 16.** Domain of Australian MCS case study showing the Top End of Australia's Northern Territory (which includes Darwin) and the Tiwi Islands. The CPOL radar location is denoted by the black triangle and its coverage by the area within the circle of dashed lines.

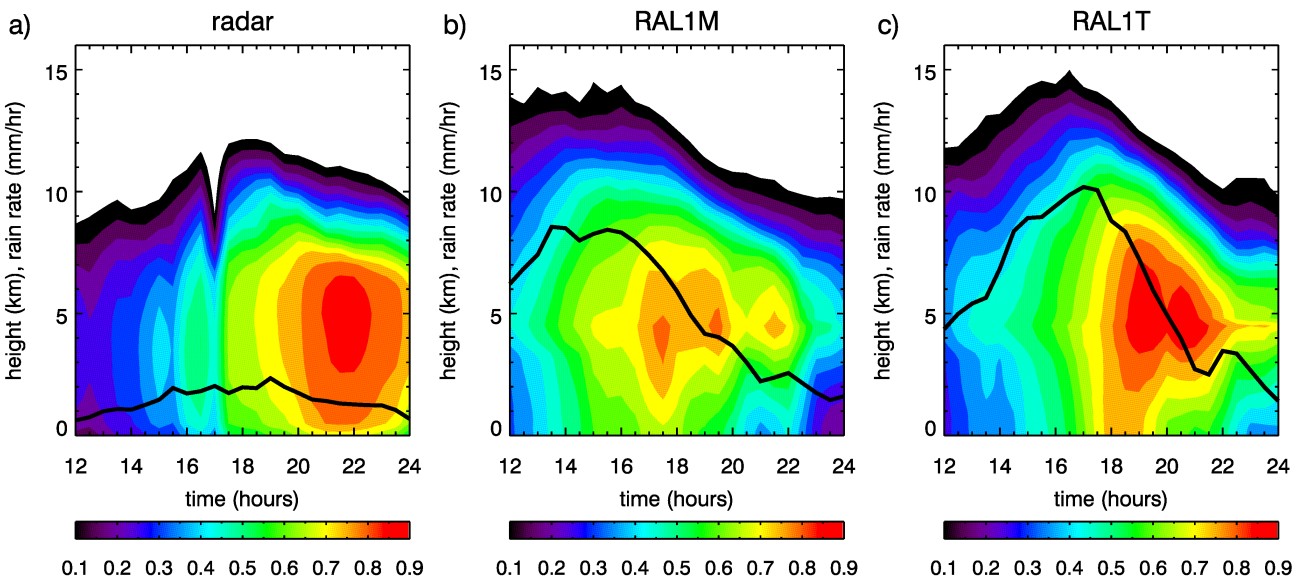

**Figure 17.** Fraction of radar area covered by reflectivities greater than 10 dBZ as a function of height and time (coloured contours) from 12:00 to 24:00 UTC on 18 February 2014. Solid lines are the time series of the domain mean rain rate (mm/hr).

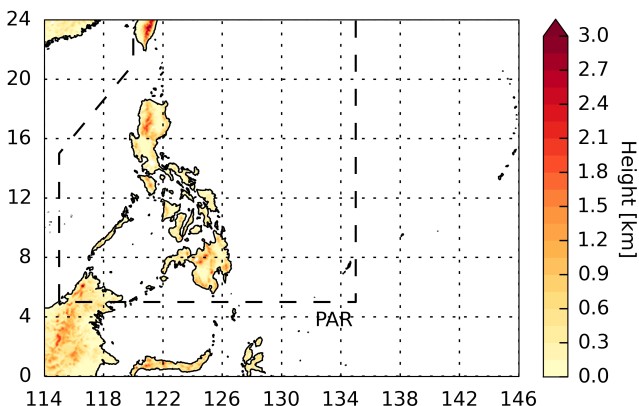

**Figure 18.** Philippines regional model domain and orography. The dashed black line shows the portion of the Philippines Area of Responsibility (PAR) inside the domain.

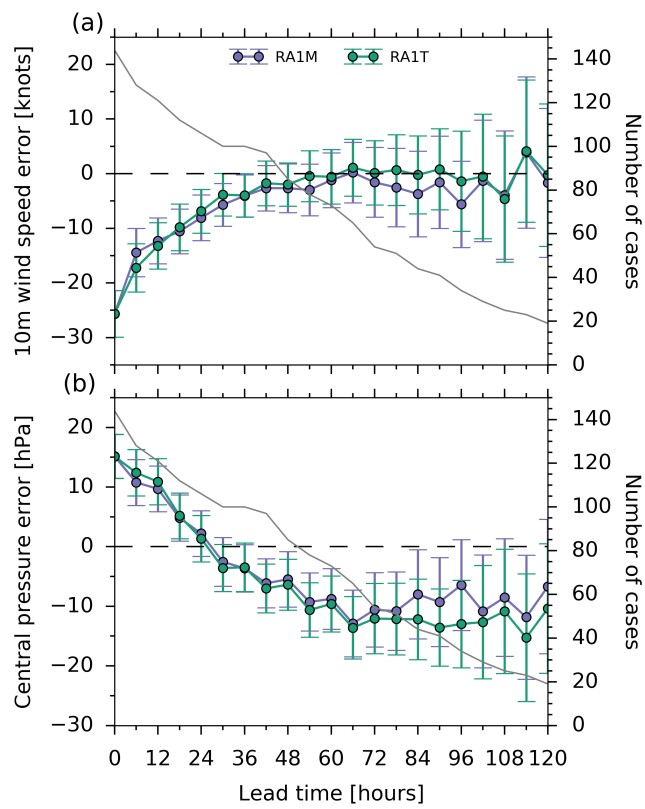

**Figure 19.** Mean bias in (a) maximum surface wind speed and (b) central pressure as a function of lead time for the RAL1-T and RAL1-M regional models. Error bars are 95% confidence intervals on the mean. The solid grey lines indicate the number of cases at each lead time (see the right-hand axis of each plot).

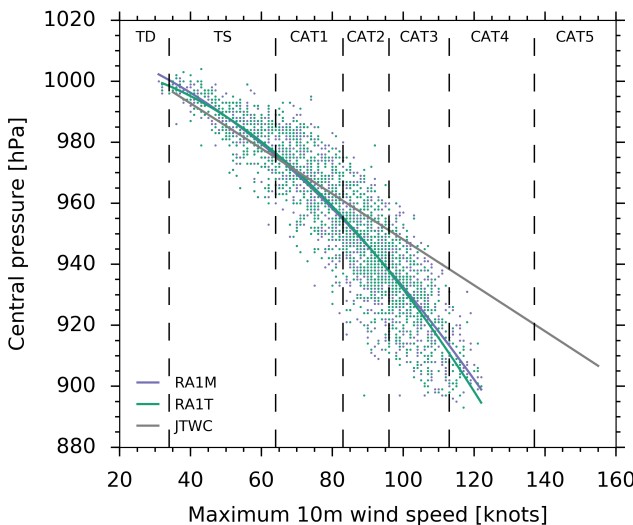

**Figure 20.** Wind-pressure relations for the RAL1-T and RAL1-M regional models. The corresponding observed relation from JTWC is shown for comparison. The solid lines are second-order polynomial fits to the data points.

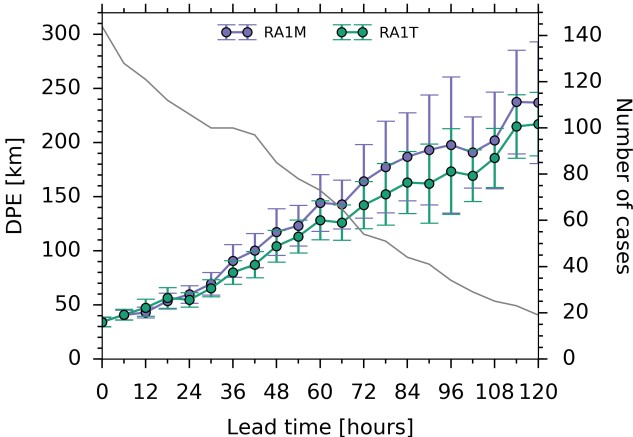

**Figure 21.** Error in forecast storm position relative to observations (direct positional error, DPE) as a function of lead time for the RAL1-T and RAL1-M models. Error bars are 95% confidence intervals on the mean. The solid grey lines indicate the number of storm cases (see the right-hand axis of the plot).

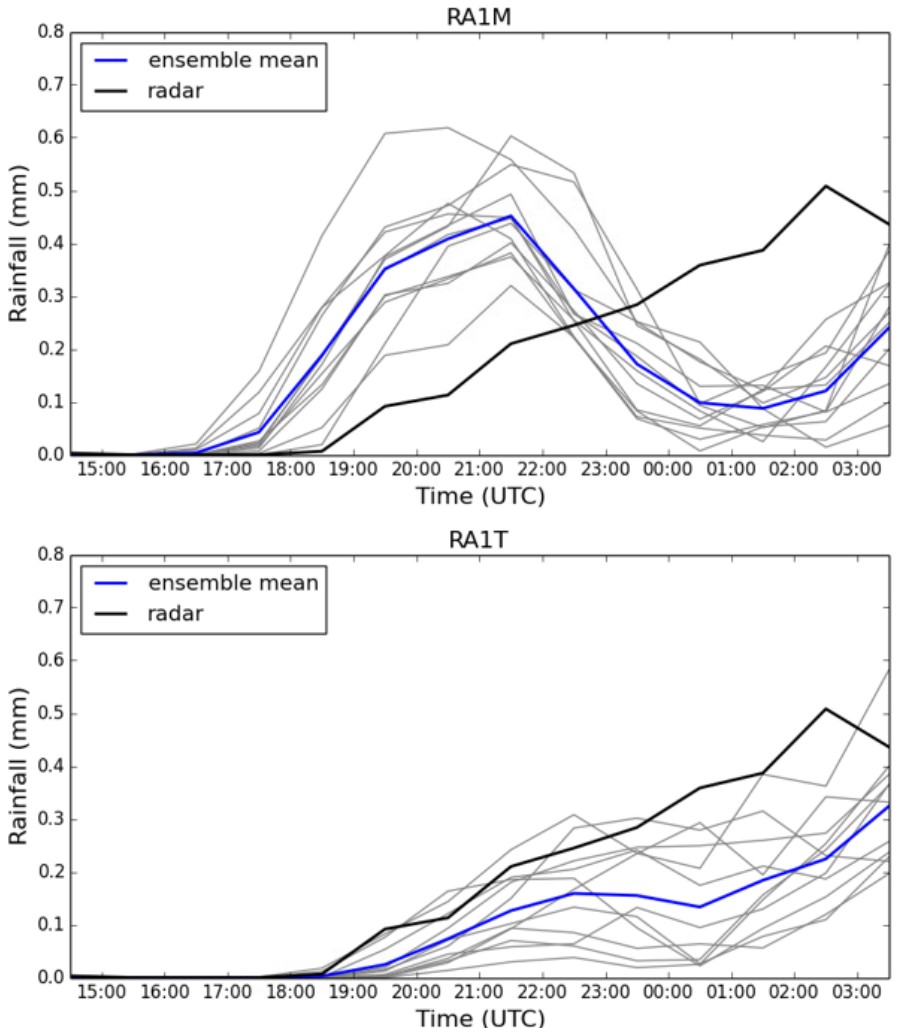

**Figure 22.** Domain-averaged precipitation from the RAL1-M ensemble (top) and RAL1-T ensemble (bottom) for 16-17 May 2017. Averaged over Texas and Oklahoma.