# Peer review of "The first Met Office Unified Model/JULES Regional Atmosphere and Land configuration, RAL1"

_Geoscientific Model Development, 2019_

## Short Comment (SC1) · 1 Jul 2019

Dear authors,

In my role as Executive editor of GMD, I would like to bring to your attention our Editorial version 1.1:

http://www.geosci-model-dev.net/8/3487/2015/gmd-8-3487-2015.html

This highlights some requirements of papers published in GMD, which is also available on the GMD website in the 'Manuscript Types' section:

http://www.geoscientific-model-development.net/submission/manuscript_types.html

In particular, please note that for your paper, the following requirements have not been

met in the Discussions paper:

- "The main paper must give the model name and version number (or other unique identifier) in the title."

- "All papers must include a section, at the end of the paper, entitled 'Code availability'. Here, either instructions for obtaining the code, or the reasons why the code is not available should be clearly stated. It is preferred for the code to be uploaded as a supplement or to be made available at a data repository with an associated DOI (digital object identifier) for the exact model version described in the paper. Alternatively, for established models, there may be an existing means of accessing the code through a particular system. In this case, there must exist a means of permanently accessing the precise model version described in the paper. In some cases, authors may prefer to put models on their own website, or to act as a point of contact for obtaining the code. Given the impermanence of websites and email addresses, this is not encouraged, and authors should consider improving the availability with a more permanent arrangement. After the paper is accepted the model archive should be updated to include a link to the GMD paper."

I understand, that you are publishing a configuration and not a model code update. However, a configuration always is intertwined with the respective model code version. Therefore please add, the respective version numbers for UM and JULES, for which the configuration RAL1 fully applies.

Additionally, I assume, that you are building your configuration on official model releases. In this case it would be sufficient to state the model code version numbers again in the code availability section. However, if your evaluation results have not been created by the use of official model versions, please make sure, to save the exact model version you applied in a permenant archive and state the information (an identifier for this archive) in the code availability section.

Yours,

Astrid Kerkweg

---

## Author Comment (AC1) · 9 Jul 2019

Both the Unified Model and JULES use a development approach in which new science is always included on a "logical switch", such that a given science configuration can be supported over a number of code versions. It is possible to run the RAL1 configuration from vn10.9 of the UM and vn5.0 of JULES onward, and support for the configuration is expected to be continued in future version releases for a number of years.

Some of these simulations were run with versions of the code that do not fully support RAL1, and hence use "subversion branches". The branches used (and their exact revision number) are defined within the individual rose suites, and permanently stored alongside the suite and the code base in the revision controlled repository.

Table 3 has been updated to show the relevant versions of the UM/JULES used in the simulations presented in the paper.

Please also note the supplement to this comment:
https://www.geosci-model-dev-discuss.net/gmd-2019-130/gmd-2019-130-AC1-supplement.pdf

**Supplement:**

**The first Met Office Unified Model/JULES Regional Atmosphere and Land configuration, RAL1**

Mike Bush[1], Tom Allen[1], Caroline Bain[1], Ian Boutle[1], John Edwards[1], Anke Finnenkoetter[1], Charmaine Franklin[2], Kirsty Hanley[1], Humphrey Lean[1], Adrian Lock[1], James Manners[1], Marion Mittermaier[1], Cyril Morcrette[1], Rachel North[1], Jon Petch[1], Chris Short[1], Simon Vosper[1], David Walters[1], Stuart Webster[1], Mark Weeks[1], Jonathan Wilkinson[1], Nigel Wood[1], and Mohamed Zerroukat[1]

[1]Met Office, FitzRoy Road, Exeter, EX1 3PB, UK
[2]Bureau of Meteorology (BoM), Melbourne, Victoria, Australia

**Correspondence:** Mike Bush (mike.bush@metoffice.gov.uk)

**Abstract.** In this paper we define the first "Regional Atmosphere and Land" (RAL) science configuration for kilometre scale modelling using the UM and JULES. "RAL1" defines the science configuration of the dynamics and physics schemes of the atmosphere and land. This configuration will provide a model baseline for any future weather or climate model developments to be described against and it is the intention that from this point forward significant changes to the system will be documented in literature. This is reproducing the process used for global configurations of the UM which was first documented as a science configuration in 2011. While it is our goal to have a single defined configuration of the model that performs effectively in all regions, this has not yet been possible. Currently we define two sub-releases, one for mid-latitudes (RAL1-M) and one for tropical regions (RAL1-T). The differences between RAL1-M and RAL1-T are documented and where appropriate, we define how the model configuration relates to the corresponding configuration of the global forecasting model.

*Copyright statement.* This work is distributed under the Creative Commons Attribution 3.0 License together with an author copyright. This license does not conflict with the regulations of the Crown Copyright.

[revised manuscript text omitted]
 (the (Brown, 1999) "standard" model is used, with $b_{LEM} = 40$, $c_{LEM} = 16$) and in the free-atmospheric mixing length (which retains RAL0's interactive one). Both give enhanced turbulent mixing in RAL1-T compared to RA1-M. The other related change is that the stochastic boundary layer perturbations are not used in RAL1-T.

A related difference is that RAL1-T has three extra prognostic fields (liquid fraction, ice fraction and mixed-phase fraction) as it uses the prognostic cloud prognostic condensate (PC2) cloud scheme (Wilson et al., 2008a). PC2 calculates sources and sinks of cloud cover and condensate and advects the updated cloud fields, hence adding some memory into the system. One advantage of PC2 over the Smith schemes is the looser coupling between variables, hence allowing a cloud to deplete its liquid water content while maintaining high cloud cover. The PC2 scheme performs better than the Smith scheme in climate simulations (Wilson et al., 2008b) and for global numerical weather prediction (Morcrette et al., 2012). It is worth noting that when run in a model using a convection scheme, the detrainment of cloud from convection is a key source of cloudiness (Morcrette and Petch, 2010; Morcrette, 2012b). When run in a model without a convection scheme (such as the RAL configuration), cloud formation from convective motions will be represented by a combination of PC2 initialization (near convective cloud base), followed by PC2 pressure forcing through the rest of the updraught. In the PC2 scheme, cloud erosion is a process that accounts for evaporation and reduction of cloud cover due to unresolved mixing near cloud edges. In the original implementation of PC2 (Wilson et al., 2008a) erosion was carried out as part of the call to the convection scheme, but in RAL1, which has no call to the convection scheme, the erosion process has been moved to occur within the microphyics scheme. In RAL1-T, the PC2 scheme is implemented as in the GA7 global model configuration (Walters et al., 2017). That is, the formulation of cloud erosion accounts for the apparent randomness of cloud fields, as described in Morcrette (2012a) and the RHcrit is calculated from the turbulent kinetic energy (Van Weverberg et al., 2016).

Another difference, particularly affecting convection in the tropics, is that the tropopause is deeper than in mid-latitudes. In order to take account of this RAL1-T uses a vertical level set labelled L80(59t;21s)38:5, which adds some additional vertical resolution in the tropical upper-troposphere at the expense of resolution in the lower boundary layer.

Figure 2 illustrates the above discussion by showing the effect of running RAL1-M and T for a case of small showers in the U.K. Unlike RAL1-M, when compared to the radar RAL1-T initiates too late and produces too large and too few showers. Table 2 contains a summary of differences between RAL1-M and RAL1-T for the convenience of the reader.

**5 Model evaluation**

In this section we apply a range of evaluation methods to demonstrate the performance of RAL1. The regional model evaluation process is rapidly evolving and has already benefitted from the multi-institutional UM partnership. The regional model is run

**Table 2.** RAL1-M and RAL1-T differences. %

[revised manuscript text omitted]

---

## Referee Comment (RC1) · Anonymous Referee #1 · 8 Aug 2019

General Comments

This paper provides a good overview of the science configurations for the regional (kilometre scale) model based on the Met Office Unified Model (UM) and JULES land surface model. With the range of locations and institutions where the model is used it is important to have a well-documented science configuration to serve as a baseline for future model development and evaluation. I found the inclusion of the evaluation over different geographical regions and weather regimes particularly interesting and useful.

Specific comments

It would be good to specify in section 1 that the Met Office you're referring to is the UK Met Office.

[Figure]

In section 2.6 where you discussed the Smith cloud scheme you give $RHt=(qv+qcl)/qsat$. Could you define qv, qcl and qsat to make it unambiguous? Also, RHc is not defined, is it RHcrit?.

The tickets discussed in Section 3 aren't sequential. It is understood that only the most important tickets are discussed and arranged per topic, but it would be useful to have a complete list of the tickets that documents all the changes to the model.

In section 3.5 "CCI" and "IGBP" is not defined. I like the description of the physical effects of the changes in the last paragraph of section 3.5 as it makes it more tangible.

The second paragraph in section 4 is a bit ambiguous. The first sentence on p15 talks about "several aspects" but then name two. Maybe the use of "such as" instead of "namely" would be better in that case? And in the following sentence about the differences in the representation of turbulence between RAL1-M and RAL1-T it would be good to specifically note that the values given there is for RA1L-T, or state the opposing values for RAL1-M as well.

GPM in section 4 and IMERG in section 5 is not defined in the text.

When you explain the "scorecard" in section 5.1, you should also indicate which direction of arrows indicate improvement/decline. Something along the lines of "triangles pointing upward (green) are indicating that model A is better than B and downward "blue" triangles indicating model B is better"

The legends in Figure 13 are not legible.

In section 5.5 it is stated that a total of 130 TC forecasts were produced, and only storm cases appearing in both RA1L-M and RAL1-T were kept. However, in Figure 19 the number of cases are up to 140 with a 0-hour lead time. Does that mean that some simulations had more than one TC at the same time? And is the mean bias for model-obs? Also, it would have been interesting to know how often storm cases appeared only in one of the experiments. Which configuration was more likely to form storms?

[Figure]

Is there a reference for the Random Parameter scheme mentioned in section 5.6?

Technical corrections

P3 line 2: "added to the RAL0 base to define RAL1-M" P7 line15: Maybe start "This represents" as a new sentence P9 line 5: "Ri is less"? P10 line 9: Remove "Therefore" P14 line 29: There is an extra "reason" P16 line 18: "most" should be "must" P17 line 25: 3-hour or 3-hourly? P17 line 32: "can lead to difficult to interpret verification scores" rather "can lead to verification scores that are difficult to interpret". P20 line 3: "reduces significantly the ability" change to "significantly reduces the ability" P23 line 5: "evaluate its performance"

---

## Referee Comment (RC2) · Anonymous Referee #2 · 19 Sep 2019

This is a nice overview of the current configuration of the UK Met Office Unified Model. I would recommend publication subject to the general and specific comments below: 1. Need to indicate that this is the UK Met Office in the title. 2. It would be nice to have a list of acronyms in an appendix. There are a lot of them! 3. Define UM and JULES at the first use (in the abstract as far as I can tell.). 4. Is there an internal report that this paper could reference? 5. While Kendon et al. 2017 is a good reference for km scale modeling, there are a lot more that could be given. General comment: There are a lot of UK Met Office references in the paper. It would make the paper more relevant if more of the communities efforts in these same areas are also referenced. Otherwise this reads as a UK Met Office report. 6. Page 3, line22. Suggest that say the "ENDGame" is the dynamical core used in the RAL1. 7. Page 3, line 14. A reference to

the hybrid height coordinate used would be appropriate here. 8. Page 6, line 12. Why has a scaling of 0.333 been applied to the Cusack et al. (1998) climatology? 9. Page 6, line 25. How is the surface albedo set? 10. Page 7, lines 14 – 23. Why isn't the CASIM microphysical parameterization developed by the Met Office not used instead of the Ballard scheme? 11. Page 8, line 7. Can you give the RH function for cloud fraction?

General Comment: I would suggest a table giving the changes from RAL0 to RAL1.

12. Page 16. I can't find Table 1 in my version of the paper. 13. Page 17. Is the size of the "triangles" in the "scorecard" proportional to a relative or absolute improvement/degradation of the model? 14. Page 18. How were the 100 cases used to verify the model chosen? 15. Page 18. What do you mean by "mixture of 00Z and 12Z runs of the Met Office global model? 16. Page 18, line 10. Suggest "By far the" instead of "One of the". 17. Page 18. Paragraph breaks are not consistent with the flow of the paper. 18. Page 18. It is interesting that the major improvement is the land surface. I wonder how dependent this is on the choice of the dates chosen to analyze? I assume there was a reason for choosing the specific 100 cases. I am wondering if this isn't biasing the results somehow. 19. Page 19. Lines 10-15. You can show anything with one case study. How robust is this result? I would suggest deleting this section. 20. Page 19. The fog "taper" is an interesting result I would like to learn more about. Why does the model produce too much fog near the surface, is dew deposition not considered? 21. Page 20, lines 13 – 16. The result described in this section needs to be expanded and explained better. 22. Page 21. Line 16. How does the storm initiation time and strength compare to observations?

General comments on figures: 1. The figures need more explanatory labels. 2. What is the verification data use for figures 4, 5,14,?

---

## Author Comment (AC2) · 24 Oct 2019

Specific comments

 c 1. It would be good to specify in section 1 that the Met Office you're referring to is the UK Met Office.

Response: "Met Office Unified Model" is the name of the model and "Met Office" is the correct name for the organization that was once known as the UK Met Office. Whilst I understand (and have some sympathy) with the point the reviewer is making, I am unable to change this. I have slightly changed the word order at the beginning of the second paragraph of the Introduction to perhaps make this more obvious.

 c 2. In section 2.6 where you discussed the Smith cloud scheme you give

RHt=(qv+qcl)/qsat. Could you define qv, qcl and qsat to make it unambiguous? Also, RHc is not defined, is it RHcrit?.

Response: I have amended the text to read: RHt=(qv+qcl)/qsat (where qv is the vapour, qcl is the liquid content and qsat is the saturation specific humidities) reaches 100% and that the grid-box only becomes overcast when RHt>=2-RHcrit.

• 3. The tickets discussed in Section 3 aren't sequential. It is understood that only the most important tickets are discussed and arranged per topic, but it would be useful to have a complete list of the tickets that documents all the changes to the model.

Response: I have added Table 6.

• 4. In section 3.5 "CCI" and "IGBP" is not defined. I like the description of the physical effects of the changes in the last paragraph of section 3.5 as it makes it more tangible.

Response: I have added a reference to table 4.

• 5. The second paragraph in section 4 is a bit ambiguous. The first sentence on p15 talks about "several aspects" but then name two. Maybe the use of "such as" instead of "namely" would be better in that case? And in the following sentence about the differences in the representation of turbulence between RAL1-M and RAL1-T it would be good to specifically note that the values given there is for RA1L-T, or state the opposing values for RAL1-M as well.

Response: I have amended the text to read: In order to cope with this, RAL1-M, has relatively weak turbulent mixing and stochastic perturbations to encourage the model fields to be less uniform and help convection initiate. If the model is run with these in the tropics the model initiates too early and convective cells tend to be too small.

Representation of turbulence (RMED tickets #12 and #26) and BL stochastic perturbations (RMED ticket #25)

[Figure]

There are two differences in the representation of turbulence between RAL1-M and RAL1-T, namely in the form of the stability functions and in the free-atmospheric mixing length. Both give enhanced turbulent mixing in RAL1-T compared to RAL1-M. RAL1-T uses the Brown (1999) "standard" model whilst RAL1-M uses the Brown (1999) "conventional" model. RAL1-T retains RAL0's interactive free-atmospheric mixing length, whilst RAL1-M uses a value of 40m. The other related change is that RAL1-T does not use the stochastic boundary layer perturbations. For more details and a summary of differences between RAL1-T and RAL1-M, see Table 2.

• 6. GPM in section 4 and IMERG in section 5 is not defined in the text.

Response: There are references to these in sections 5 and 5.1

• 7. When you explain the "scorecard" in section 5.1, you should also indicate which direction of arrows indicate improvement/decline. Something along the lines of "triangles pointing upward (green) are indicating that model A is better than B and downward "blue" triangles indicating model B is better" showing whether the model version being tested is better or worse than a previous incarnation.

Response: I have amended the text to read: Triangles pointing upward (green) indicate that the test model is better than the control and downward (purple) triangles indicate the control model is better.

• 8. The legends in Figure 13 are not legible.

Response: Figure 13 has had its legends made legible.

• 9. In section 5.5 it is stated that a total of 130 TC forecasts were produced, and only storm cases appearing in both RA1L-M and RAL1-T were kept. However, in Figure 19 the number of cases are up to 140 with a 0-hour lead time. Does that mean that some simulations had more than one TC at the same time?

Is the mean bias for model-obs?

Also, it would have been interesting to know how often storm cases appeared only in one of the experiments. Which configuration was more likely to form storms?

Response: The answer to the first two points is yes and I have slightly amended the text. 130 TC forecasts were run. When the storm sample was homogenised across the two experiments (RAL1M/T), there were 126 initialisation times remaining, i.e. some forecasts were discarded because either no storm was found in either experiment, or a storm was found in one experiment but not the other. This sort of thing can happen if the storm is weak and drops below one of the pre-defined thresholds used in the tracker code.Âă Âă In Fig 19 the number of cases at T+0 is 144. This is because there were 18 forecasts where two storms were present in the domain at the initialisation time (and 126 + 18 = 144). In 9 of these forecasts TCs Chan-Hom and Linfa were both present at T+0, and in the other 9 it was TCs Koppu and Champi.

Yes the mean bias is for model-obs.

The TC verification software we use (see Heming 2017 for details, reference below) only tracks and verifies storms that were observed to exist at the model analysis time so we do not have any statistics regarding TC genesis at present. In principle this is something we could look at in the future, but we suspect there would be little difference in genesis statistics between RAL1M/T. Âă Heming, J. T. (2017), Tropical cyclone tracking and verification techniques for Met Office numerical weather prediction models. Met. Apps, 24: 1-8. doi:10.1002/met.1599Âă Âă • 10. Is there a reference for the Random Parameter scheme mentioned in section 5.6? Response: Added McCabe et al. (2016).

• Technical corrections

P3 line 2: "added to the RAL0 base to define RAL1-M" P7 line15: Maybe start "This represents" as a new sentence. P9 line 5: "Ri is less"? P10 line 9: Remove "Therefore" P14 line 29: There is an extra "reason" P16 line 18: "most" should be "must" P17 line 25: 3-hour or 3-hourly? P17 line 32: "can lead to difficult to interpret verification

scores" rather "can lead to verification scores that are difficult to interpret". P20 line 3: "reduces significantly the ability" change to "significantly reduces the ability" P23 line 5: "evaluate its performance"

Response: All technical corrections have been made.

Please also note the supplement to this comment: https://www.geosci-model-dev-discuss.net/gmd-2019-130/gmd-2019-130-AC2-supplement.pdf

**Supplement:**

**The first Met Office Unified Model/JULES Regional Atmosphere and Land configuration, RAL1**

Mike Bush[1], Tom Allen[1], Caroline Bain[1], Ian Boutle[1], John Edwards[1], Anke Finnenkoetter[1], Charmaine Franklin[2], Kirsty Hanley[1], Humphrey Lean[1], Adrian Lock[1], James Manners[1], Marion Mittermaier[1], Cyril Morcrette[1], Rachel North[1], Jon Petch[1], Chris Short[1], Simon Vosper[1], David Walters[1], Stuart Webster[1], Mark Weeks[1], Jonathan Wilkinson[1], Nigel Wood[1], and Mohamed Zerroukat[1]

[1]Met Office, FitzRoy Road, Exeter, EX1 3PB, UK
[2]Bureau of Meteorology (BoM), Melbourne, Victoria, Australia
**Correspondence:** Mike Bush (mike.bush@metoffice.gov.uk)

**Abstract.** In this paper we define the first "Regional Atmosphere and Land" (RAL) science configuration for kilometre scale modelling using the Unified Model (UM) as the basis for the atmosphere and the Joint UK Land Environment Simulator (JULES) for the land. "RAL1" defines the science configuration of the dynamics and physics schemes of the atmosphere and land. This configuration will provide a model baseline for any future weather or climate model developments to be described against and it is the intention that from this point forward significant changes to the system will be documented in literature. This is reproducing the process used for global configurations of the UM which was first documented as a science configuration in 2011. While it is our goal to have a single defined configuration of the model that performs effectively in all regions, this has not yet been possible. Currently we define two sub-releases, one for mid-latitudes (RAL1-M) and one for tropical regions (RAL1-T). The differences between RAL1-M and RAL1-T are documented and where appropriate, we define how the model configuration relates to the corresponding configuration of the global forecasting model.

*Copyright statement.* This work is distributed under the Creative Commons Attribution 3.0 License together with an author copyright. This license does not conflict with the regulations of the Crown Copyright.

[revised manuscript text omitted]

---

## Author Comment (AC3) · 25 Oct 2019

1. Need to indicate that this is the UK Met Office in the title.

Response: "Met Office Unified Model" is the name of the model and "Met Office" is the correct name for the organization that was once known as the UK Met Office. Whilst I understand (and have some sympathy) with the point the reviewer is making, I am unable to change this. I have slightly changed the word order at the beginning of the second paragraph of the Introduction to perhaps make the UK more obvious. Please note that the analogous Global Atmosphere series of papers in GMD (Walters et al, 2011, 2014, 2017) also have a title beginning "The Met Office Unified Model Global Atmosphere..."

[Figure]

2. It would be nice to have a list of acronyms in an appendix. There are a lot of them!

Response: I have added Table 5 containing a list of acronyms.

3. Define UM and JULES at the first use (in the abstract as far as I can tell.).

Response: I have modified the abstract to read: In this paper we define the first "Regional Atmosphere and Land" (RAL) science configuration for kilometre scale modelling using the Unified Model (UM) as the basis for the atmosphere and the Joint UK Land Environment Simulator (JULES) for the land.

4. Is there an internal report that this paper could reference?

Response: Nothing that is of direct relevance.

5. While Kendon et al. 2017 is a good reference for km scale modeling, there are a lot more that could be given. General comment: There are a lot of UK Met Office references in the paper. It would make the paper more relevant if more of the communities efforts in these same areas are also referenced. Otherwise this reads as a UK Met Office report.

Response: I have added three more relevant references (Baldauf et al. (2011), Brousseau et al. (2016) and Bengtsson et al. (2017)).

6. Page3, line22. Suggest that say the "ENDGame" is the dynamical core used in the RAL1.

Response: This section deals with RAL0. I have amended the text to say that RAL0 uses the UM's ENDGame dynamical core. Also the new Table 5 explicitly notes that RAL1 also uses the ENDGame dynamical core.

7. Page3, line14. A reference to the hybrid height coordinate used would be appropriate here.

Response: I have amended the text to say: "A terrain-following hybrid height coordinate

is used that it is a mix of both pure height (i.e. flat levels) and terrain following levels (Davies et al., 2005)."

8. Page 6, line 12. Why has a scaling of 0.333 been applied to the Cusack et al. (1998) climatology?

Response: I have amended the text to say "and the contribution from dust has been scaled by 0.3333 compared to the original climatology of Cusack (1998) as the dust loading of the basic climatology over land (which includes arid areas) is too high for the UK."

9. Page 6, line 25. How is the surface albedo set?

Response: I have amended the text to say "The emissivity and the albedo of the surface are set by the JULES land surface model (see Section 2.8). A single frequency-averaged emissivity is specified for each surface type (see Walters et al. (2014) for the numerical values). For the surface albedo, the radiative transfer in plant canopies uses the two-stream radiation scheme and spectral parameters of Sellers (1985)."

10. Page 7, lines 14 – 23. Why isn't the CASIM microphysical parameterization developed by the Met Office not used instead of the Ballard scheme?

Response: CASIM is being actively developed and the intention is that it will be incorporated into a future RAL version. At the time that RAL1 was released, the code wasn't yet ready for operational use because it was slower and didn't contain the necessary coupling to the boundary layer scheme for forecasting of foggy cases (essential for the UK in the winter). Both of these issues are in the process of being addressed. We haven't mentioned CASIM in the paper as we want the manuscript to contain details of parametrizations included in RAL1 rather than digressing into details of parametrizations which were not included, potentially confusing the reader.

11. Page 8, line 7. Can you give the RH function for cloud fraction?

Response: The equation is given below. I don't feel it is necessary to include it in

the paper as it is quite long and doesn't really add much. However if the editor feels strongly that we should include it, then we can.

In the Smith scheme, the cloud fraction c is calculated from qn using: if qn < -1 c=0; elseif qn<0 c=0.5*((1+qn)^2); elseif qn<1 c=1-(0.5*((1-qn)^2)); else c=1; end

where qn=(rht-1)/(1-rhc) and rht=(qv+qcl)/qsat.

When using the EACF we use the same formulation but instead of c being calculated from qn it is calculated from qn' where qn'=(qn+0.184)/(1-0.184); where the 0.184 has been determined from trying to better match the observations of Wood and Field and while also improving model performance.

General comment: I would suggest a table giving the changes from RAL0 to RAL1. Response: I have added Table 6.

12. Page 16. I can't find Table 1 in my version of the paper.

Response: Table 1 (model timesteps) is at the bottom of page 4. Table 2 (M&T diffs) is at the top of page 16.

13. Page 17. Is the size of the "triangles" in the "scorecard" proportional to a relative or absolute improvement/degradation of the model?

Response: I have amended the text to say: "The area of the triangles is proportional to the absolute improvement (or deterioration) of the model and the triangles are outlined in black if the change is statistically significant, at the 0.05 level determined using the Wilcoxon signed-rank test."

14. Page 18. How were the 100 cases used to verify the model chosen?

Response: I have significantly added to the text to say: "The UK evaluation consisted of a hierarchy of testing. Firstly, individual science changes (RMED tickets) were tested by running 100 case studies with a 1.5km horizontal grid-length, using the same domain as the Operational UKV model (Figure 3). These were simple downscaling runs

(from the Met Office Global model) with no data assimilation. The cases sampled a wide range of meteorological conditions from the period July 2014 to April 2017 and comprised roughly equal numbers from each season. The cases were a mixture of poor forecasts (as identified by forecasters), high impact weather and normal everyday weather. The verification results from this stage of testing were used in the decision making process of whether individual science changes were performing well enough to progress to the next round of testing. Secondly, the tickets were packaged up into a "proto-RAL1" package and the same case study tests repeated. Typically there may be several "proto" packages trialled before a preferred package is chosen. Thirdly to test the impact of including data assimilation in RAL1, one month long UKV 3D-VAR Data Assimilation trials were run for Summer and Winter 2016. The exact choice of dates for the case studies (and indeed the data assimilation trials) can obviously affect the results, but the reason for running the case studies is to provide a relatively cheap and quick test of model changes before moving on to the more expensive data assimilation trials."

15. Page 18. What do you mean by "mixture of 00Z and 12Z runs of the Met Office global model?

Response: See response to 14.

16. Page 18, line 10. Suggest "By far the" instead of "One of the".

Response: I have amended the text to read "By far the most significant improvement in RAL1 is the surface temperature".

17. Page 18. Paragraph breaks are not consistent with the flow of the paper.

Response: I have slightly changed the paragraph breaks.

18. Page 18. It is interesting that the major improvement is the land surface. I wonder how dependent this is on the choice of the dates chosen to analyze? I assume there was a reason for choosing the specific 100 cases. I am wondering if this isn't biasing

the results somehow.

Response: See response to 14. I have also added some more text: "Figure 4 shows the HiRA scorecard comparing RAL1 performance with RAL0 for the 100 case studies and Figure 5 shows the results for the 3D-Var Winter and Summer trials. The first thing to note is that there is remarkably good agreement between the case study and the 3D-Var trial results. This shows that the case studies can give a good indication of likely performance in data assimilation trials and that the exact choice of dates is not crucial to the results provided enough cases are run."

19. Page 19. Lines 10-15. You can show anything with one case study. How robust is this result? I would suggest deleting this section.

Response: I have decided to keep this section, but have amended the text to read: "Overall RAL1 shows statistically significant degradation to cloud fraction RPS at most lead times in both case study (Figure 4) and 3D-Var Winter trials (Figure 5 left panel). Subjective assessment of RAL1 by forecasters found that whilst largely very similar to RAL0, RAL1 tends to break up lower cloud faster than RAL0, especially where that cloud is fragmented. Whilst on average the reduction in cloud amounts verifies worse, in some cases it is good. Figure 13 shows a stratocumulus case from 23rd June 2015."

20. Page 19. The fog "taper" is an interesting result I would like to learn more about. Why does the model produce too much fog near the surface, is dew deposition not considered?

Response: I have amended the text to read: "RAL1 reduces the optical depth of fog as a result of the droplet taper change (ticket 1) and further discussion of fog processes and model performance can be found in Boutle et al. (2018)."

There definitely is dew deposition in the model, although fog is very sensitive to exactly how this is represented, hence the importance of the drop number in the taper since that governs the cloud droplet sedimentation rate onto the surface.

21. Page 20, lines 13 – 16. The result described in this section needs to be expanded and explained better.

Response: I have expanded the text significantly to read: "The impact of PC2 is to increase light rain amounts and decrease very heavy rain amounts compared to the Smith scheme. Effectively this makes the model more dissipative and this leads to a reduction of small-scale structure which enables the large-scale envelope of features like Sumatran Squalls to be better handled and hence to propagate more realistically. The increased free-atmospheric mixing further increases the dissipation, and the two together were found to improve the ability of the model to propagate Sumatran Squalls faster and further, rather than have them not develop or dissipate prematurely."

22. Page 21. Line 16. How does the storm initiation time and strength compare to observations?

Response: Figure 19 answers the strength compared to observations question. As for the storm initiation time (when genesis occurs), the TC verification software we use (see Heming 2017 for details, reference below) only tracks and verifies storms that were observed to exist at the model analysis time so we do not have any statistics regarding TC genesis at present. In principle this is something we could look at in the future, but we suspect there would be little difference in genesis statistics between RAL1M/T.

Heming, J. T. (2017), Tropical cyclone tracking and verification techniques for Met Office numerical weather prediction models. Met. Apps, 24: 1-8. doi:10.1002/met.1599

General comments on figures: 1. The figures need more explanatory labels. 2. What is the verification data used for figures 4, 5,14,?

Response: Added that "HiRA uses synoptic observations (see section 5) to the captions for figures 4 and 5. Figure 14 already says that 3 hourly GPM is used.

[Figure]

Please also note the supplement to this comment:
https://www.geosci-model-dev-discuss.net/gmd-2019-130/gmd-2019-130-AC3-supplement.pdf

—————————————————————

[Figure]

**Supplement:**

**The first Met Office Unified Model/JULES Regional Atmosphere and Land configuration, RAL1**

Mike Bush[1], Tom Allen[1], Caroline Bain[1], Ian Boutle[1], John Edwards[1], Anke Finnenkoetter[1], Charmaine Franklin[2], Kirsty Hanley[1], Humphrey Lean[1], Adrian Lock[1], James Manners[1], Marion Mittermaier[1], Cyril Morcrette[1], Rachel North[1], Jon Petch[1], Chris Short[1], Simon Vosper[1], David Walters[1], Stuart Webster[1], Mark Weeks[1], Jonathan Wilkinson[1], Nigel Wood[1], and Mohamed Zerroukat[1]

[1]Met Office, FitzRoy Road, Exeter, EX1 3PB, UK
[2]Bureau of Meteorology (BoM), Melbourne, Victoria, Australia

**Correspondence:** Mike Bush (mike.bush@metoffice.gov.uk)

**Abstract.** In this paper we define the first "Regional Atmosphere and Land" (RAL) science configuration for kilometre scale modelling using the Unified Model (UM) as the basis for the atmosphere and the Joint UK Land Environment Simulator (JULES) for the land. "RAL1" defines the science configuration of the dynamics and physics schemes of the atmosphere and land. This configuration will provide a model baseline for any future weather or climate model developments to be described against and it is the intention that from this point forward significant changes to the system will be documented in literature.
5    This is reproducing the process used for global configurations of the UM which was first documented as a science configuration in 2011. While it is our goal to have a single defined configuration of the model that performs effectively in all regions, this has not yet been possible. Currently we define two sub-releases, one for mid-latitudes (RAL1-M) and one for tropical regions (RAL1-T). The differences between RAL1-M and RAL1-T are documented and where appropriate, we define how the model
10    configuration relates to the corresponding configuration of the global forecasting model.

*Copyright statement.* This work is distributed under the Creative Commons Attribution 3.0 License together with an author copyright. This license does not conflict with the regulations of the Crown Copyright.

[revised manuscript text omitted]